Corrected: Publisher correction

# Z-ring membrane anchors associate with cell wall synthases to initiate bacterial cell division

Manuel Pazos[1], Katharina Peters[1], Mercedes Casanova[2], Pilar Palacios[2], Michael VanNieuwenhze [3], Eefjan Breukink [4], Miguel Vicente[2] & Waldemar Vollmer[1]

During the transition from elongation to septation, *Escherichia coli* establishes a ring-like peptidoglycan growth zone at the future division site. This preseptal peptidoglycan synthesis does not require the cell division-specific peptidoglycan transpeptidase PBP3 or most of the other cell division proteins, but it does require FtsZ, its membrane-anchor ZipA and at least one of the bi-functional transglycosylase-transpeptidases, PBP1A or PBP1B. Here we show that PBP1A and PBP1B interact with ZipA and localise to preseptal sites in cells with inhibited PBP3. ZipA stimulates the glycosyltransferase activity of PBP1A. The membrane-anchored cell division protein FtsN localises at preseptal sites and stimulates both activities of PBP1B. Genes *zipA* and *ftsN* can be individually deleted in *ftsA\** mutant cells, but the simultaneous depletion of both proteins is lethal and cells do not establish preseptal sites. Our data support a model according to which ZipA and FtsN-FtsA have semi-redundant roles in connecting the cytosolic FtsZ ring with the membrane-anchored peptidoglycan synthases during the pre-septal phase of envelope growth.

---

[1] Centre for Bacterial Cell Biology, Institute for Cell and Molecular Biosciences, Newcastle University, Richardson Road, Newcastle upon Tyne NE2 4AX, UK. [2] Centro Nacional de Biotecnología-Consejo Superior de Investigaciones Científicas (CNB-CSIC), Darwin 3, 28049 Madrid, Spain. [3] Molecular and Cellular Biochemistry Department, Biology Department, Indiana University, 212S. Hawthorne Dr, Bloomington, IN 47405, USA. [4] Membrane Biochemistry and Biophysics, Bijvoet Center for Biomolecular Research, Department of Chemistry, Faculty of Science, Utrecht University, Padualaan 8, 3584 CH Utrecht, The Netherlands. Correspondence and requests for materials should be addressed to W.V. (email: w.vollmer@ncl.ac.uk)

Most bacteria contain a peptidoglycan (PG) sacculus to counteract the osmotic pressure and maintain the shape of the cell[1]. During the cell cycle, the sacculus is enlarged and remodelled to facilitate cell growth and division. In *Escherichia coli* this process is achieved by dynamic multi-enzyme complexes, the elongasome and divisome, involved in elongation or septation, respectively, which are anchored to the cytoplasmic membrane. The periplasmic steps of PG synthesis are catalysed by glycosyltransferases (GTases), which polymerise the lipid II substrate into glycan strands, and transpeptidases (TPases) that cross-link the peptides of adjacent strands[2]. PBP2 and PBP3 are essential TPases involved in cell elongation and division, respectively[3–5]. The bifunctional synthases PBP1A and PBP1B (encoding both GTase and TPase activities) have semi-redundant roles in cell elongation and division[6]. PBP1B interacts with PBP3 and is enriched at division sites[7]. PBP1A interacts with PBP2 and affects cell diameter[8], suggesting a role in elongation. However, single-molecule tracking of fluorescent PBP1A fusion proteins revealed slow and fast moving molecules with different trajectories than the essential cell elongation proteins PBP2 and RodA[9,10]. The activities of the PG synthases are coordinated or regulated by outer membrane lipoproteins (LpoA and LpoB) and components of the divisome and elongasome, the SEDS proteins (RodA and FtsW) and bacterial cytoskeletal proteins (MreBCD and FtsZ)[2,11–13].

The synthesis of the septal PG at mid-cell is controlled by the divisome complex, the components of which span from the cytosol to the outer membrane. At early stages of cell division FtsZ forms a cytosolic ring-like structure (Z-ring) that is anchored to the inner membrane by ZipA and FtsA[14,15]. This proto-ring complex[16] serves as a scaffold to hierarchically recruit the other components of the divisome including FtsK, FtsQLB, FtsW, FtsI (PBP3) and FtsN[11]. FtsN was originally reported as the last essential protein recruited to division site[17] but recent studies showed that a portion of FtsN is also recruited at early stages through a cytosolic interaction with FtsA[18,19]. The main septal PG synthases PBP1B and PBP3 interact with each other[7] and with different components of the divisome such as FtsN and FtsW, which regulate their synthetic activities[13,20]. The integral membrane protein FtsW flips lipid II[21] and lacks GTase activity[13] in the test tube. However, other groups proposed that FtsW and other members of the SEDS proteins have GTase activity[22,23], as has been shown for RodA from *B. subtilis*[22], a RodA-PBP2 fusion from *E. coli*[24] and FtsW from several species[25]. FtsN also interacts with PBP1A[20], consistent with the observation that PBP1A is able to bypass the absence of PBP1B during septal PG synthesis. In fact, there is evidence for interactions between components of the elongasome and divisome during the transition from cell elongation to division[26–28].

During this transition new PG is synthesised at mid-cell before any visible constriction, causing cell elongation in a ring-like growth zone at the future division (preseptal) site. This PG is inserted in a PBP3-independent manner, as it takes place in the presence of the PBP3-inhibitor aztreonam, and therefore was named PIPS for *P*BP3-(or *P*enicillin-) *I*ndependent *P*eptidoglycan *S*ynthesis[29–31]. For the purpose of this study, we refer to PIPS as preseptal PG synthesis. Preseptal PG synthesis can be readily observed in cells upon long-term in situ labelling of PG with ᴅ-cysteine followed by growth in aztreonam-containing medium, isolation of sacculi and visualisation of ᴅ-cysteine residues, and appears as label-free zones at future division sites[30,31]. In *E. coli* preseptal synthesis has a relatively small contribution to the total length growth of the cell[30], but *Caulobacter crescentus* elongates significantly using this mode of growth[32]. So far, in *E. coli* only FtsZ, ZipA and either PBP1A or PBP1B are described as essential proteins for preseptal PG synthesis, and several proteins from

both elongasome and divisome complexes are not required, e.g. RodA, FtsA, FtsEX, FtsK or FtsQ[31]. However, even though these proteins or downstream cell division proteins are not required for preseptal PG synthesis, they might still localise at these sites.

The minimal requirement of cell division proteins for preseptal PG synthesis might suggest that ZipA acts as a linker between the cytosolic Z-ring and the periplasmic PG synthases. ZipA is dispensable in cells containing certain point mutations in *ftsA* (named *ftsA**)[33–35] and preseptal PG synthesis takes also place in these strains[31], indicating that a possible role of ZipA in linking the Z-ring and the PBPs could be bypassed in the *ftsA** background.

In this work we show that ZipA interacts with both, PBP1A and PBP1B, linking the cytosolic Z-ring with the PG synthases. Also FtsN localises at preseptal sites and both, FtsN and ZipA stimulate PBP1A and PBP1B (albeit differently), implying roles of FtsN and ZipA in preseptal PG synthesis. Our observation of the synthetic lethality of *zipA* and *ftsN* in a *ftsA** mutant strain and the drastic decrease of preseptal PG synthesis bands during *zipA* and *ftsN* depletion supports a model according to which ZipA and FtsN (the latter bound to FtsA) have redundant functions in linking the Z-ring and the PBPs during preseptal PG synthesis.

## Results

**ZipA interacts with PBP1A, PBP1B and PBP3 and not with FtsN.** We hypothesised that PG synthases are guided by cytoskeletal elements during preseptal PG synthesis, as they are during cell elongation and division, and that this is achieved by interactions between the relatively few proteins essential for the process. ZipA anchors FtsZ to the membrane via its transmembrane region and would be ideally positioned to connect the cytosolic Z-ring with the membrane-anchored PBPs required for the synthesis of new PG in the periplasm (Fig. 1a). We tested if ZipA interacts in vivo with PBP1A and PBP1B, and with the late cell division proteins PBP3 and FtsN. Using specific antibodies in a cellular cross-linking and co-immunoprecipitation assay we detected interactions between ZipA and PBP1A, PBP1B and PBP3, but not with FtsN (Fig. 1b). To elucidate if these interactions are direct, we purified the proteins with or without an oligohistidine tag and performed pulldown assays with nickel-nitrilotriacetic acid (Ni-NTA) agarose. ZipA interacted directly with the PBPs as untagged PBP1A and PBP1B were retained on the Ni-NTA beads only in the presence of His-ZipA (Fig. 1c, d), and the untagged ZipA was retained only in the presence of His-PBP3 (Fig. 1e). A soluble version of ZipA lacking its first 25 amino acid residues showed decreased interaction with any of the PBPs (Fig. 1c–e), suggesting that the transmembrane and/or adjacent regions of ZipA are involved in these interactions. We next replaced the transmembrane region of ZipA by the unrelated WALP23 peptide, which forms a single transmembrane alpha helix formed by repeats of alternating leucine and alanine residues, flanked by two tryptophan residues at the N- and C-termini, and has been previously studied as an artificial transmembrane peptide[36]. Purified WALP23-sZipA did not interact with PBP1A, PBP1B or PBP3 (Supplementary Fig. 1c and 1d), showing that the interaction of ZipA with PBPs requires the native amino acid sequence of its transmembrane region and does not occur with just any transmembrane helix. In these pulldown assays we also observed an interaction between ZipA and FtsN. However unlike the interactions between ZipA and PBP1A or PBP1B, the ZipA–FtsN interaction was not detected at higher concentration of detergent (Supplementary Fig. 1). Hence, the ZipA–FtsN interaction might be disrupted by hydrophobic molecules such as detergents and phospholipids, which would be consistent with the inability to cross-link both proteins in the cell (Fig. 1b). All together these results suggest that ZipA interacts with PBPs and that it might link the Z-ring with the PG synthases.

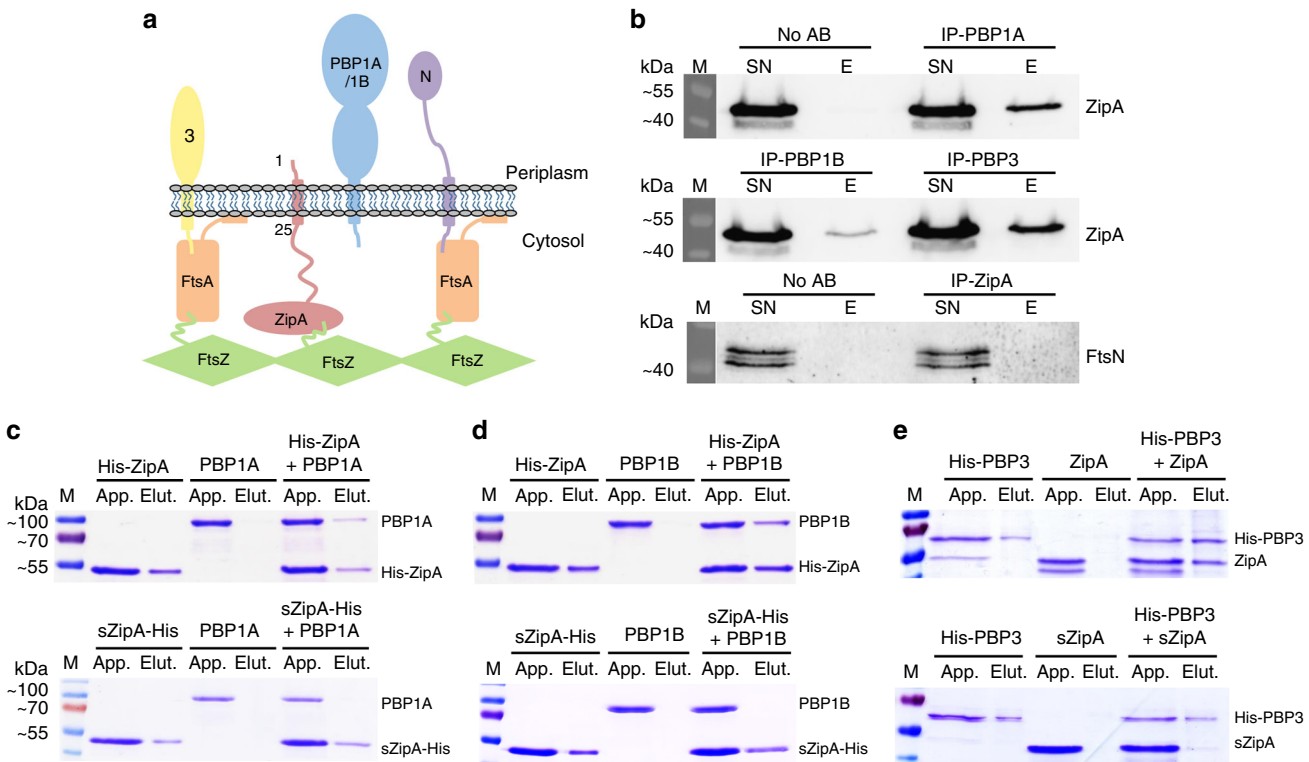

**Fig. 1** ZipA interacts directly with PBP1A, PBP1B and PBP3 when its periplasmic and transmembrane regions are present. **a** Schematic representation of proteins essential for preseptal PG synthesis and the FtsA-PBP3 (3) and FtsA-FtsN (N) interactions. The residues defining the N-terminal periplasmic and transmembrane regions of ZipA are labelled. **b** Wild-type cells were cross-linked by the addition of DSP, co-immunoprecipitated using specific antibodies recognising PBP1A, PBP1B, PBP3 or ZipA, and bound to G-agarose beads. The cross-linking was reversed by the addition of β-mercaptoethanol, the proteins were separated by SDS-PAGE and the interacting proteins, ZipA or FtsN, were immunodetected by western blot using specific antibodies. SN unbound proteins, E elution. Uncropped scans of the western blots are supplied as Supplementary Fig. 9. **c-e** Interactions assayed using purified proteins in a cross-linking/pulldown experiment in the presence of 0.05% Triton X-100, in which His-ZipA or sZipA-His, a soluble variant lacking the first 25 residues (periplasmic and transmembrane regions), were incubated with either PBP1A (**c**) or PBP1B (**d**), and His-PBP3 was incubated with either ZipA or sZipA (**e**). In all the cases, the histidine tagged proteins were able to retain the untagged protein when ZipA contained the periplasmic and transmembrane regions. The soluble version of ZipA did not retain either PBP1A or PBP1B, and it was not retained by His-PBP3. Uncropped pictures of the gels are supplied as Supplementary Fig. 10. M molecular weight markers, kDa kilodalton, App. applied sample, Elut. sample eluted from Ni-NTA beads

**PBP1A, PBP1B, ZipA and FtsN localise at preseptal positions.** To test if PBP1A and PBP1B are recruited to preseptal positions in the cell, we localised both PBPs in exponentially growing cells using specific antibodies before and after the inhibition of cell division by aztreonam. The effect of aztreonam on cell growth and division was monitored by measuring the optical density and the increase in particle counts (Supplementary Fig. 2). Before the addition of aztreonam, both PBPs localised along the lateral cell periphery and in case of PBP1B also at cell division site (Fig. 2 and Supplementary Fig. 3), confirming previous localisation data[7,8]. Interestingly, the localisation pattern of PBP1A and PBP1B changed when cells grew in the presence of aztreonam. Forty minutes after the addition of the antibiotic, both PBPs localised mainly at potential cell division sites where they may be available to participate in preseptal PG synthesis (Fig. 2 and Supplementary Fig. 4). We also determined the localisation of FtsN, as we hypothesised that it might be recruited to preseptal positions based on its known interaction with FtsA, PBP1A and PBP1B[18,20]. FtsN was present at preseptal or cell division sites and along the cell membrane in the presence or absence of aztreonam (Fig. 2 and Supplementary Fig. 4). As control we also localised FtsZ and ZipA, which both showed the expected localisation at cell division site (in the absence of aztreonam) and at potential cell division sites (in the presence of aztreonam) (Fig. 2 and Supplementary Fig. 4). We also localised the above-

mentioned proteins in cells lacking PBP1A (ΔmrcA) or PBP1B (ΔmrcB) to test if the observed localisation patterns depend on the presence of either PBP1A or PBP1B, as it is known that both proteins can perform preseptal PG synthesis. Nearly all the examined cells showed rings containing FtsN (>77%), FtsZ (>92%) or ZipA (>95%) irrespective of the time of incubation with aztreonam. The number of rings with associated PBP1A (>77%) or PBP1B (>92%) decreased slightly upon incubation with the antibiotic (by 10% or 20%, respectively) (Supplementary Fig. 3 and Supplementary Table 3). The normal localisation of FtsZ, ZipA and FtsN in cells lacking one of the PBPs might be expected because each of them is capable of compensating for the absence of the other. These data suggest that PBP1A and PBP1B are recruited to preseptal positions in the cell, presumably linked to the Z-ring by ZipA, although FtsN and other cell division proteins may also contribute to their recruitment.

To test if ZipA is required for the localisation of PBP1A and PBP1B at preseptal positions, both PBPs were immunolocalised in ZipA-depleted cells. As previously described[37] the depletion of ZipA was not complete but the amount of ZipA was sufficiently reduced to cause filamentation of cells. In these cells, the residual ZipA localised along the cell length (Supplementary Fig. 5). PBP1A and PBP1B mirrored ZipA localisation along the length of the filamented cells (Supplementary Fig. 5), in contrast to the preseptal localisation of ZipA in aztreonam-treated WT cells

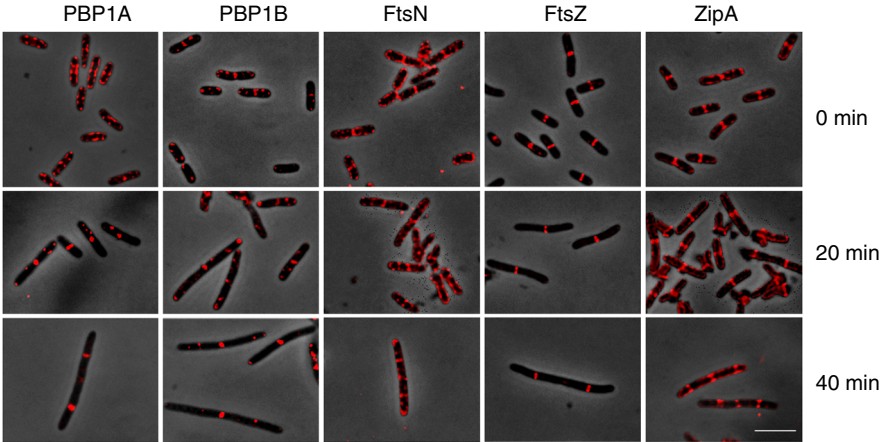

**Fig. 2** PBP1A, PBP1B, FtsN, FtsZ and ZipA localise at potential cell division sites in the presence of aztreonam. Merged micrographs of phase contrast and immunolocalisation images of PBP1A, PBP1B, FtsN, FtsZ or ZipA, detected with purified antibodies, in BW25113 cells grown in the absence or presence of 0.3 μg mL$^{-1}$ aztreonam (for 20 or 40 min). Scale bar represents 5 μm

(Fig. 2). FtsA and FtsZ-rings localised at preseptal sites in ZipA-depleted cells as expected (Supplementary Fig. 5)[14]. These data suggest that ZipA is required for the correct localisation of PBP1A and PBP1B at preseptal positions.

**Effects of ZipA and FtsN on the GTase activity of PBPs.** To further characterise the interactions between ZipA and PBPs, we tested the effect of ZipA on the activities of PBP1A and PBP1B. Measuring the consumption of fluorescently labelled lipid II we observed a 2.6 ± 0.2-fold increase in the GTase rate of PBP1A and a 1.4 ± 0.1-fold increase in the GTase rate of PBP1B GTase by ZipA (Fig. 3a, b, left panels; Supplementary Fig. 6). A soluble version of ZipA (sZipA) had weaker or no effect on the GTase activity of PBP1A or PBP1B. FtsN slightly increased the activity of PBP1A (1.3 ± 0.2-fold) and, as published previously[38], stimulated PBP1B 2.6 ± 0.4-fold. For PBP1A the effects of ZipA and FtsN were additive, yielding a 3.4 ± 0.5-fold stimulation of PBP1A, whereas for PBP1B the stimulation was 3.0 ± 0.7-fold. PBP3 did not significantly affect the stimulation of PBPs by ZipA and FtsN. To quantify the TPase activity we determined the percent of peptides present in cross-links in an endpoint assay using radiolabelled lipid II substrate. ZipA alone or in combination with FtsN, PBP3, or both proteins, did not significantly alter the percentage of cross-links produced by PBP1A or PBP1B (Fig. 3a,b, right panels). These results suggest that the interactions of ZipA and FtsN with either PBP1A or PBP1B stimulate the GTase activity of each protein by compatible mechanisms.

**Stimulation of PBP1A and PBP1B at low concentration.** PBP1B self-interacts and shows poor GTase and TPase activities at concentrations significantly below the $K_D$ of dimerisation, i.e. when it adopts a monomeric state. At this condition, FtsN increases both the GTase and the TPase activities suggesting that it might stabilise the more active dimeric form of PBP1B[20,39]. We therefore tested the effects of interacting proteins on PBP1A and PBP1B, using the endpoint PG synthesis assay with PBP1A and PBP1B present at low concentration. The reaction products were separated by high-pressure liquid chromatography (HPLC). At low enzyme concentration PBP1A was poorly active, and it was not significantly stimulated by the addition of ZipA, FtsN, PBP3 or its specific activator LpoA (Fig. 4a). The lack of stimulation by LpoA was expected under these conditions since PBP1A alone has neglectable GTase activity and LpoA only stimulates the TPase activity which depends on ongoing GTase reactions[40,41].

In case of PBP1B, the presence of ZipA increased the monomeric GTase product peak (Penta, compound **2**) in comparison to the reaction without ZipA (Fig. 4b). Both samples contained very low amount of the cross-linked GTase/TPase product peak (TetraPenta; compound **3**), suggesting poor cross-linking activity. When PBP3 or FtsN was added to PBP1B, lipid II was quantitatively consumed and the Penta and TetraPenta products were found (Fig. 4b), showing stimulation of PBP1B and confirming the previous data for the effect of FtsN[20]. The absence of the transmembrane region of FtsN (sFtsN) significantly decreased the stimulation of PBP1B (Supplementary Fig. 7b) consistent with previously published data[20]. As expected, PBP1B's specific activator LpoB alone or in combination with PBP3 or ZipA also stimulated both activities of PBP1B (Fig. 4b). The structures of the main reaction products is shown in Fig. 4c. As expected, ZipA, FtsN or PBP3 alone were inactive (Supplementary Fig. 7a), which excludes the presence of a contaminating PG synthase in these protein preparations.

We next assessed the GTase activity at low concentration of PBP1A or PBP1B in the presence of ampicillin using a fluorescent labelled lipid II and separating by sodium dodecyl sulfate polyacrylamide gel electrophoresis (SDS-PAGE) the glycan chains produced (Fig. 4d, e). In the case of PBP1A, glycan chains were barely detected consistent with the data from the endpoint assay with radioactive lipid II (Fig. 4d). Glycan chains could be detected by increasing the contrast of the image. The quantity and length distribution of the glycan chains produced by PBP1A was not altered by ZipA, FtsN, PBP3, LpoA or combinations of these proteins (Fig. 4d). PBP1B had poor GTase activity at low concentration producing almost undetectable glycan chains (Fig. 4e). Consistent with the previous findings (Fig. 4b) PBP1B produced significantly more glycan chains in the presence of PBP3, LpoB or FtsN, and GTase activity was moderately increased in the presence of ZipA (Fig. 4e). The highest activity, as judged from the almost complete consumption of lipid II, was obtained in the presence of LpoB and either ZipA or PBP3. The control protein bovine serum albumin (BSA) did not stimulate PBP1B at low concentration, excluding the possibility that proteins in general unspecifically stimulate the synthase at this condition. Our analysis also showed that PBP1B produced more of the longer glycan chains (with more than 20 disaccharide units) in the presence of FtsN, which were not produced in the presence of soluble version of FtsN (sFtsN) (Supplementary Fig. 7c); the glycan chains produced in the presence of LpoB, PBP3 or ZipA were shorter. As expected, ZipA, PBP3, FtsN or

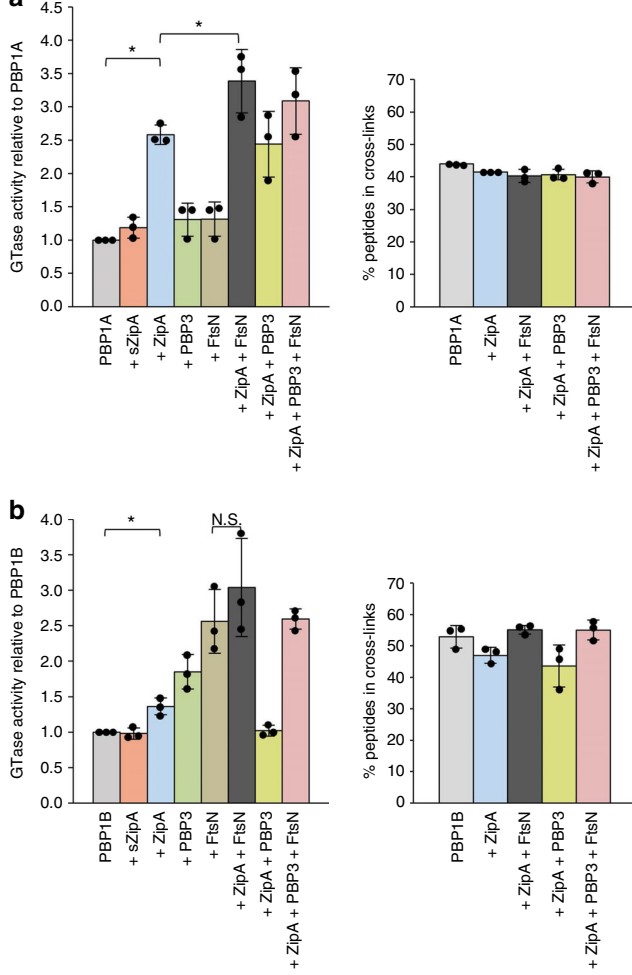

**Fig. 3** ZipA stimulates the GTase but not TPase activity of PBP1A and PBP1B. Synthetic activities of PBP1A (**a**) and PBP1B (**b**) in the presence of different interacting partners. GTase activity (left panels) was measured as the consumption of dansylated lipid II substrate by PBP1A or PBP1B. The GTase rates are shown as mean ± s.d. of three independent experiments, after normalisation to PBP1A or PBP1B alone values. TPase activities (right panels) were determined by the percentage of peptides in cross-links present in newly synthesised PG by PBP1A or PBP1B using radiolabelled lipid II substrate, and they are shown as mean ± s.d. of three independent experiments. The corresponding dot plots (filled circles) are overlaid in each bar chart. Student's *t*-test (two-tailed) was used for statistical analysis (N.S. not significant; *$P < 0.05$)

sFtsN alone were inactive in this assay excluding any possible contamination by a GTase (Supplementary Fig. 7c).

Overall, our activity assays show that ZipA moderately stimulates the GTase activity of PBP1A and PBP1B, probably via a different mechanism than FtsN, which also stimulates the TPase activity of PBP1B. FtsN, PBP3 and LpoB might stabilise the more active dimeric form of PBP1B[20].

**ZipA and FtsN have different but synergistic roles.** Although ZipA is essential for preseptal PG synthesis, certain point mutations in *ftsA* (called *ftsA**) allow the cells to perform preseptal PG synthesis in the absence of *zipA*. FtsA* versions appear to be impaired in self-interaction, resulting in an earlier or more efficient recruitment of PBP3 and FtsN by FtsA* monomers[34,42]. Because PBP3 is not required for preseptal PG synthesis, we hypothesised that the interaction between FtsA* and FtsN, and

hence FtsN itself, could be essential for preseptal cell wall synthesis in the absence of *zipA*. To test this hypothesis we visualised the incorporation of a fluorescent D-amino acid (HADA)[43] into single and double *zipA* and *ftsN* conditional mutants (both genes encoded in thermosensitive replication plasmids) in an *ftsA** (*ftsA*^E124A) background strain upon addition of aztreonam to generate preseptal PG growth zones. FtsA* mutant cells grew in the absence of ZipA or FtsN (although the colony formation was not fully restored in the absence of FtsN[44,45]) but were not viable in the absence of both (Fig. 5a) demonstrating that each protein becomes essential in the absence of the other. After 1.5 h at non-permissive temperature PG was labelled by the incorporation of HADA for 30 min and the excess of dye was removed prior to the inhibition of septation for 40 min by addition of aztreonam. The parental and single depletion strains showed preseptal PG synthesis, as demonstrated by the presence of non-labelled zones at potential cell division sites (Fig. 5d–f). In the case of the *zipA ftsN* double mutant there was a drastic decrease in localised insertion of preseptal PG, leading to a continuously labelled side wall (Fig. 5g). FtsZ-rings were present in these filamentous cells indicating that the lack of preseptal PG synthesis was not due to mislocalised FtsZ (Supplementary Fig. 8). Quantification of preseptal PG synthesis zones per cell length unit revealed an ~80% decrease in the double mutant strain compared to the WM2935 parental strain (Fig. 5c and Supplementary Table 1). Similar results were obtained for an *ftsA** Δ*zipA* Δ*ftsN* mutant strain containing only the *zipA*-encoding plasmid (Supplementary Table 1). Importantly, the cells were still elongating as demonstrated by the brighter signal at cell poles, indicating PG incorporation into the side wall during the chase period, and the increase in the optical density of the culture during the time period (1.5–3 h at non-permissive temperature) of the experiment (Fig. 5b). These data indicate that FtsN and ZipA are both able to support preseptal PG synthesis which, however, does not take place in the simultaneous absence of both proteins.

## Discussion

In this work we show that ZipA interacts with PBP1A, PBP1B and PBP3 through the transmembrane region, but not with FtsN. The transmembrane region of ZipA is essential for its full functioning, and more specific roles have been suggested for it in addition to anchoring the protein to the inner membrane[46]. These roles might include the specific interaction with other divisome components such as the PG synthases. Our results support a role for ZipA as a linker between the cytosolic Z-ring and the periplasmic PG synthases. Our localisation data suggest that these interactions occur during preseptal PG synthesis although we cannot exclude that they also occur during later stages of cell division. It also suggests that the absence of preseptal PG synthesis in ZipA-depleted cells is due to the mislocalisation of PBP1A and PBP1B.

Most of the FtsA* versions that compensate the absence of *zipA* show reduced self-interaction, suggesting that FtsA monomers recruit late cell division proteins such as PBP3 and FtsN more efficiently than FtsA oligomers[34,35,47]. ZipA could be needed to disrupt FtsA oligomers to produce sufficient amount of active monomers, explaining why ZipA becomes dispensable in cells expressing FtsA*. Some of the FtsA* mutants (including FtsA^E124A) can also bypass the absence of other downstream cell division proteins such as FtsN or FtsK, but the *ftsN* gene could not be deleted in a *ftsA** Δ*zipA* strain[35]. This suggests that ZipA must have an essential role that overlaps with the role of FtsN. Our results support a model according to which an overlapping essential role of ZipA and FtsN is the interaction with the PG

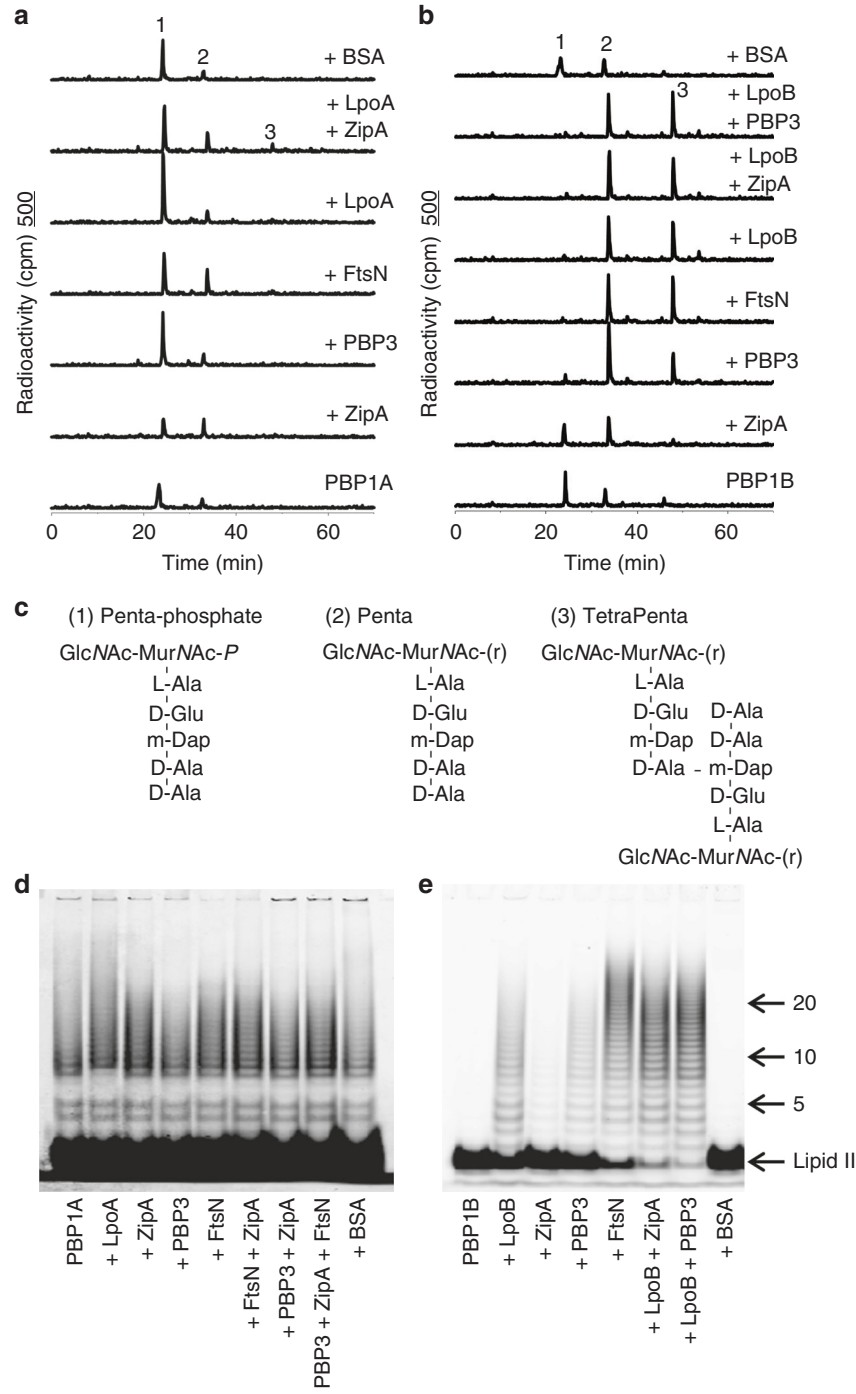

**Fig. 4** Stimulation of PBP1A and PBP1B activities at low concentration. **a**, **b** Representative HPLC chromatograms of PBP1A (**a**) and PBP1B (**b**) in vitro PG synthesis reactions in the presence of the proteins indicated, using radioactive lipid II as substrate. The synthesised PG was digested with cellosyl, reduced with sodium borohydride and analysed by HPLC. Peak 1 is generated from glycan chain ends and unreacted lipid II, peak 2 is a GTase product and peak 3 is a GTase/TPase product. (**c**) Structures of the main products of the in vitro synthesis reactions. **d**, **e** SDS-PAGE analysis of glycan chains synthesised by PBP1A (**d**) and PBP1B (**e**) GTase activity at low concentration. Reactions were incubated at 37 °C for 1 h, using a mixture of unlabelled and ATTO(550)-labelled lipid II as substrate, in the presence of the indicated interacting proteins. The numbers refer to disaccharide units. The contrast of the image in panel **d** was increased due to the low signal observed

synthases initially participating in preseptal PG synthesis (PBP1A or PBP1B) and later on in septation (PBP1A or PBP1B, and PBP3)[7,13,20,48]. Indeed, ZipA and FtsA, presumably bound to FtsN, localise at the constriction site until septation is completed[49], suggesting that this linker function extends from preseptal PG synthesis into septum synthesis. The FtsA–FtsN

interaction has recently been shown to promote the recruitment of late cell division proteins and is essential to bypass the absence of FtsEX and FtsK[50].

Our model (Fig. 6) proposes that the Z-ring and PG synthases are linked at early stages of cell division by ZipA and FtsA-FtsN, although the latter linker requires the presence of ZipA to

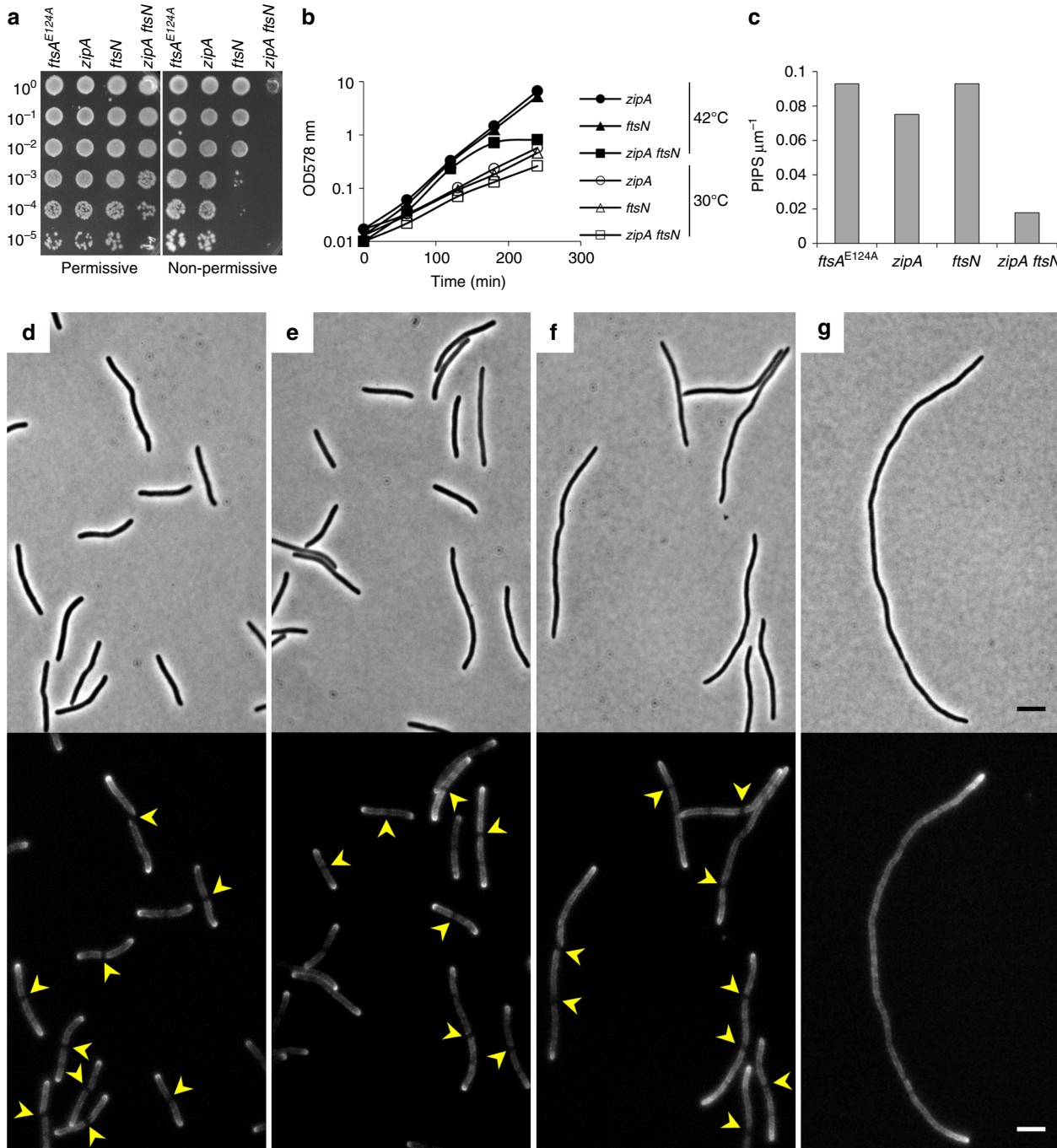

**Fig. 5** FtsN and ZipA are required for preseptal peptidoglycan synthesis. **a** Ten-fold serial dilutions spotted plate of the parental strain WM2935 (ftsA$^{E124A}$), MPW22 (ftsA$^{E124A}$ ΔzipA/rep$^{ts}$ zipA), MPW23 (ftsA$^{E124A}$ ΔftsN/rep$^{ts}$ ftsN) and MPW29 (ftsA$^{E124A}$ ΔzipA ΔftsN//rep$^{ts}$ zipA ftsN) strain cultures incubated at 30 °C (permissive) and 42 °C (non-permissive). **b** Representative growth curves of strains under permissive (open symbols) and non-permissive temperature (close symbols). Cell culture optical density (578 nm) was kept below 0.4 during the whole experiment. **c** Quantification of preseptal peptidoglycan synthesis (PIPS) bands per unit of cell length in the mentioned strains under non-permissive conditions (42 °C). **d–g** Phase contrast and HADA fluorescent images for detection of preseptal PG synthesis in WM2935 (**d**), MPW22 (**e**), MPW23 (**f**) and MPW29 (**g**) cells under non-permissive conditions (42 °C). The non-fluorescently labelled preseptal PG synthesis bands are indicated by yellow arrows. Scale bars represent 5 μm

promote FtsA monomerization[34]. The ftsA* allele increases the number of FtsA* monomers and, hence, the number of FtsA*-FtsN complexes available for interactions with the PBPs, rendering ZipA dispensable. If zipA is absent, FtsN becomes the only linker between the cytosolic proto-ring (formed by FtsA* and FtsZ) and the bifunctional PG synthases, explaining the lack of viability of the ftsA* cells in the absence of both, ZipA and FtsN. A similar defect in the transition from cell elongation to cell division has been proposed to occur when the direct interaction between MreB and FtsZ is impaired, causing cell filamentation and loss of viability[26]. Although mature cell division rings (containing PBP3 and FtsN among other proteins) were located at the division sites, in this case there was no preseptal or septal PG synthesis, suggesting a defective transfer of the PG synthesis machinery from the elongasome and the divisome[26]. Although other cell division proteins (FtsK, FtsQ, FtsEX, PBP3) are

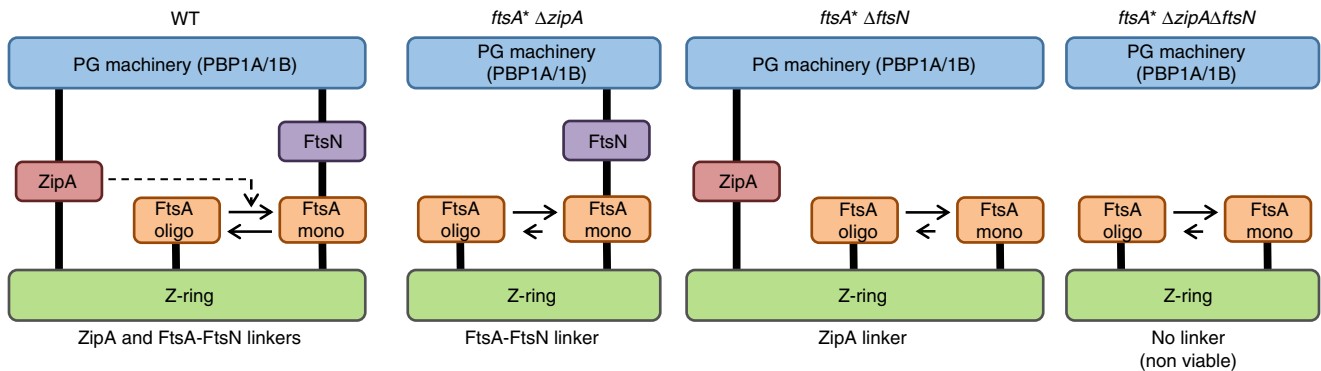

**Fig. 6** Model for preseptal PG synthesis in the presence or absence of ZipA. FtsZ, ZipA and either PBP1A or PBP1B are essential for preseptal PG synthesis. The cytosolic FtsZ ring (Z-ring) and the PG synthases are linked at early stages of cell division by ZipA and by FtsA-FtsN. ZipA promotes monomerization of FtsA, which is needed to obtain the FtsA–FtsN complex. In the presence of FtsA* (FtsA mutants impaired in dimerisation) the interaction between FtsA* and FtsN is enhanced, rendering ZipA dispensable and FtsN essential to link the Z-ring and the PG synthesis machinery. In the absence of FtsN, ZipA becomes the only linker between both cytosolic and periplasmic proteins. When both ZipA and FtsN are absent the link is lost, leading to non-viable cells. The cell membrane is not shown in this diagram

dispensable for preseptal PG synthesis[31] and become essential only later in cell division[13,45,51–53], we cannot exclude that they localise at preseptal positions and affect the activities of PBP1A or PBP1B. The recently proposed GTase FtsW is unlikely to be required and present at preseptal positions because neither FtsK nor FtsQ are required for preseptal PG synthesis[31], and it is known that FtsK and the FtsQLB complex are necessary for recruiting FtsW to mid-cell[54–56].

In addition to the role of ZipA and FtsN in the recruitment of the PG synthases to mid-cell, our results show that both proteins have a positive effect on the activities of PBP1A and PBP1B. At high concentration of PBP1A and PBP1B, both of their GTase rates were modestly increased by ZipA, and in case of PBP1A there was a synergistic effect with FtsN. The cross-linking of the PG produced was unaffected. This suggests that ZipA and FtsN exert their stimulatory effects by compatible mechanisms. At low concentrations of PBP1A and PBP1B (below the $K_D$ of dimerisation) their activities were significantly decreased. Previous work suggests that PBP1B monomers are less active than dimers and that the stimulation of both GTase and TPase activities by FtsN may be due to the stabilisation of PBP1B dimers[20]. It remains to be tested if dimer stabilisation is the reason for the effect of FtsN and other proteins that stimulate the enzyme at low concentration. PBP1B interacts with a number of proteins including PBP3 (ref.[7]), FtsW[13], FtsN[20], LpoB[40,57,58], CpoB[59], MipA[60], PgpB[61] and now ZipA. It is unlikely that all PBP1B molecules in the cell interact with all of these proteins at all times. Hence, it remains to be determined when and how precisely these interactions contribute to controlling the activities and functioning of PBP1B in the cell.

A functional connection between ZipA and FtsN has been suggested due to their presence in γ-proteobacteria[62] and because of the synthetic lethality of the double mutant in an ftsA* background strain[35]. Our data are consistent with a synergistic and semi-redundant role of both proteins in acting as linkers between the cytosolic Z-ring and the PG synthases during preseptal PG synthesis. We hypothesise that redundancy in linker function contributes to the robustness of the process, similar to the redundancies found for other PG-related functions[63].

## Methods

**Strains, plasmids and growth conditions**. *E. coli* strains and plasmids used are listed in Supplementary Table 2. Standard genetic methods including P1 phage transduction and transformation were used for strain construction.

*Strains*: To produce strain MPW22 [ftsA^E124A zipA::aph/pCH32 (zipA, ftsZ)], WM2935 (ftsA^E124A) was transformed with pCH32 (zipA, ftsZ) and transduced with the P1 lysate from the strain WM1304 (ΔzipA::aph). Strain MPW23 [ftsA^E124A ftsN::cat/pWM2245 (ftsN)] was constructed using WM2935 (ftsA^E124A) cells transformed with pWM2245 (ftsN) and transduced with the P1 lysate from the strain WM2355 (ΔftsN::cat). To produce strain MPW29 [ftsA^E124A zipA::aph ftsN::cat/pCH32 (zipA, ftsZ) pWM2245 (ftsN)], cells from MPW22 [ftsA^E124A zipA::aph/pCH32 (zipA, ftsZ)] were transformed with pWM2245 (ftsN) and transduced with the P1 lysate from the strain WM2355 (ΔftsN::cat). Strain MPW30 [ftsA^E124A zipA::aph ftsN::cat/pCH32 (zipA, ftsZ)] was constructed by transduction of MPW22 [ftsA^E124A zipA::aph/pCH32 (zipA, ftsZ)] cells with the P1 lysate from the strain WM2355 (ΔftsN::cat). Mutations were verified by PCR. The depletion of the proteins of interest was quantified by western blot using specific antibodies.

*Plasmids*: Plasmid pPZW05 was generated by two PCR fragments: an amplified product obtained from pET28a(+) and the oligonucleotides FwZipAHis-V (5′-TTA GTT CCT CGT GGT TCT CTC GAG CAC CAC CAC CAC CAC-3′) and RvZipAHis-V (5′-CAA ATC CTG CAT CAT CCA TGG TAT ATC TCC TTC TTA AAG-3′), and an amplified product obtained from BW25113 genomic DNA and the oligonucleotides FwZipAHis-I (5′-GAA GGA GAT ATA CCA TGG ATG ATG CAG GAT TTG CGT CTG-3′) and RvZipAHis-I (5′-AGA ACC ACG AGG AAC TAA GGC GTT GGC GTC TTT GAC TTC-3′). Same volumes of each PCR fragments were mixed, heated to 98 °C and cooled down to room temperature. The DNA mix was digested with DpnI and transformed into DH5α competent cells. pPZW06 was obtained by the same procedure, using pET28a(+) and the oligonucleotides FwZipAHis-V (5′-TTA GTT CCT CGT GGT TCT CTC GAG CAC CAC CAC CAC CAC-3′) and RvsolZipAHis-V (5′-ACG GCT GGT CCA CAT CCA TGG TAT ATC TCC TTC TTA AAG-3′), and BW25113 genomic DNA and the oligonucleotides FwsolZipAHis-I (5′-GAA GGA GAT ATA CCA TGG ATG TGG ACC AGC CGT AAA GAA C-3′) and RvZipAHis-I (5′-AGA ACC ACG AGG AAC TAA GGC GTT GGC GTC TTT GAC TTC-3′). pPZW22 was obtained by the same procedure, using pPZW06 and the oligonucleotides FwZipAHis-V (5′-TTA GTT CCT CGT GGT TCT CTC GAG CAC CAC CAC CAC CAC-3′) and RvWALPZipA-V (5′-CAG GGC GAG AGC AAG CGC AAT GC CAA GGC TAA CCA CCA AGC CAT CCA TGG TAT ATC TCC TTC TTA AAG-3′), and pPZW06 and the oligonucleotides FwWALPZipA-I (5′-GCG CTT GCT CTC GCC CTG GCA TTG GCG TTA GCC CTG TGG TGG GCG TGG ACC AGC CGT AAA GAA CGA TC-3′) and RvZipAHis-I (5′-AGA ACC ACG AGG AAC TAA GGC GTT GGC GTC TTT GAC TTC-3′). In this construct the WALP23 peptide (MAWWLALALALALALALALALWWA) was fused to the N-ter end of the soluble ZipA. Constructs were confirmed by DNA sequencing the genes of interest.

*Growth of E. coli cells*: Unless stated otherwise, cells were grown in Miller Luria-Bertani (LB) medium (1% tryptone, 0.5% yeast extract, 1% NaCl) for protein production. Cells of temperature-sensitive (ts) strains were grown at 30 °C (permissive condition) or 42 °C (non-permissive condition) in Lennox LB medium (1% tryptone, 0.5% yeast extract, 0.5% NaCl). When appropriate, antibiotics were supplied to the media (100 μg mL⁻¹ ampicillin, 10 μg mL⁻¹ chloramphenicol, 50 μg mL⁻¹ kanamycin, 50 μg mL⁻¹ spectinomycin). For in vivo co-immunoprecipitation assays cells were grown in Lennox LB medium (Fischer Scientific) at 37 °C.

**Protein purification**. The following proteins were purified following published protocols: PBP1B[7], LpoA(sol)[64], LpoB(sol)[57], FtsN-His[38], His-PBP3[38], FtsN-His (sol)[38] and PBP1A[65].

*PBP1A*: LOBSTR cells containing plasmid pTK1Ahis were grown in autoinduction medium (Miller LB medium supplemented with 0.5% glycerol, 0.05% glucose and 0.2% α-lactose) supplemented with ampicillin for 18 h at 30 °C. Cells were harvested by centrifugation (6200 × *g*, 15 min, 4 °C) and the pellet was resuspended in buffer I (25 mM Tris/HCl, 100 mM NaCl, 10 mM MgCl$_2$, pH 7.5). After addition of 200 μM phenylmethylsulfonylfluoride (PMSF), 1 in 1000 dilution of protease inhibitor cocktail (Sigma-Aldrich) and DNase, the cells were disrupted by sonication (Branson digital). The cell lysate was centrifuged (130,000 × *g*, 60 min, 4 °C) and the supernatant was discarded. The membrane pellet was resuspended in buffer II (25 mM Tris/HCl, 1 M NaCl, 10 mM MgCl$_2$, 10% glycerol, pH 7.5) during 3 h at 4 °C, and centrifuged (130,000 × *g*, 60 min, 4 °C). The supernatant was discarded and the membrane pellet was resuspended in extraction buffer (25 mM Tris/HCl, 1 M NaCl, 2% Triton X-100 reduced (Sigma-Aldrich), 10 mM MgCl$_2$, 10% glycerol, pH 7.5) and incubated overnight with mixing at 4 °C. Resuspended sample was centrifuged (130,000 × *g*, 60 min, 4 °C) and the supernatant was incubated with 5 mL of Ni-NTA Superflow (Qiagen) during 2 h at 4 °C with gentle stirring, which had been pre-equilibrated in extraction buffer. The resin was poured into a gravity column and washed with 10 volumes of wash buffer (25 mM Tris/HCl, 1 M NaCl, 0.2% Triton X-100 reduced, 10% glycerol, 10 mM MgCl$_2$, 20 mM imidazole, pH 7.5). Bound protein was eluted with elution buffer (25 mM Tris/HCl, 1 M NaCl, 0.2% Triton X-100 reduced, 10 mM MgCl$_2$, 500 mM imidazole, pH 7.5). About 2 U mL$^{-1}$ of restriction grade thrombin (Merck Millipore) were added to the Ni-NTA eluted protein to remove the oligohistidine tag during dialysis against 3 L of dialysis buffer I (25 mM Tris/HCl, 1 M NaCl, 10% glycerol, 10 mM EGTA, 10 mM MgCl$_2$, pH 7.5) for 20 h at 4 °C. Sample was dialysed against 3 L of dialysis buffer II (10 mM sodium acetate, 500 mM NaCl, 10 mM MgCl$_2$, 10% glycerol, pH 4.8) for 4 h at 4 °C, and 3 L of dialysis buffer III (10 mM sodium acetate, 300 mM NaCl, 10 mM MgCl$_2$, 10% glycerol, pH 4.8) for 18 h at 4 °C. The sample was diluted 1:1 with buffer IV (10 mM sodium acetate, 10 mM MgCl$_2$, 10% glycerol, 0.2% Triton X-100 reduced, pH 4.8) and applied in buffer AKTA A (10 mM sodium acetate, 150 mM NaCl, 10 mM MgCl$_2$, 0.2% Triton X-100 reduced, pH 4.8) to a 5 mL HiTrap SP HP column using an ÄKTA Prime (GE Healthcare Bio-Sciences) for cation exchange chromatography (flow rate 1 mL min$^{-1}$). The protein eluted in a gradient from 150 mM to 1 M NaCl. Protein-containing fractions were dialysed against storage buffer (25 mM HEPES/NaOH, 500 mM NaCl, 10 mM MgCl$_2$, 10% glycerol, pH 7.5) and stored at −80 °C.

*PBP1B*: His-PBP1B was purified from strain BL21(DE3) pDML924 grown in 1.5 L of LB supplemented with kanamycin at 30 °C to an OD$_{578}$ of 0.4–0.6. Overproduction was induced by adding 1 mM of IPTG and incubating the cells for 3 h at 30 °C. Cells were harvested by centrifugation (10,000 × *g*, 15 min, 4 °C) and the pellet resuspended in 80 mL of buffer I (25 mM Tris/HCl, 500 mM NaCl, 1 mM EGTA, 10% glycerol, pH 7.5) to which 1 in 1000 protease inhibitor cocktail, 100 μM PMSF and DNase were added. The resuspension was frozen at −80 °C until required (<3 months), at which time it was rapidly thawed and cells disrupted by sonication. The membrane fraction was pelleted by centrifugation at 130,000 × *g* for 1 h at 4 °C and resuspended in buffer II (25 mM Tris/HCl, 5 mM MgCl$_2$, 1 M NaCl, 20% glycerol, 2% Triton X-100, pH 7.5) with protease inhibitor cocktail and PMSF added as before. Extracted membranes were again centrifuged at 130,000 × *g* for 1 h at 4 °C to remove remaining insoluble debris before 1:1 dilution with buffer III (25 mM Tris/HCl, 5 mM MgCl$_2$, 1 M NaCl, 40 mM imidazole, 20% glycerol, pH 7.5) and application to an equilibrated 5 mL HiTrap column attached to an ÄKTA Prime system with fraction collection. Once the sample had been fully applied the column was washed with 40 mL of buffer IV (25 mM Tris/HCl, 5 mM MgCl$_2$, 1 M NaCl, 40 mM imidazole, 20% glycerol, 0.2% Triton X-100, pH 7.5). Bound His-PBP1B was eluted stepwise with buffer V (25 mM Tris/HCl, 5 mM MgCl$_2$, 1 M NaCl, 400 mM imidazole, 20% glycerol, 0.2% Triton X-100, pH 7.5). His-PBP1B containing fractions were pooled into a regenerated cellulose dialysis membrane with a molecular weight cut-off of 6–8 kDa (Spectrum Labs) and treated with 2 U mL$^{-1}$ of thrombin (Novagen) for 20 h at 4 °C during dialysis against dialysis buffer I (25 mM Tris/HCl, 5 mM MgCl$_2$, 1 M NaCl, 20% glycerol, pH 7.5). Protein was then dialysed in preparation for ion exchange chromatography, first against dialysis buffer II (20 mM sodium acetate, 1 M NaCl, 10% glycerol, pH 5.0); then against dialysis buffer II with 300 mM NaCl; and finally against dialysis buffer II with 100 mM NaCl prior to application to an equilibrated 1 mL HiTrap SP HP column attached to an ÄKTA Prime system with fraction collection. The column was equilibrated in buffer A (20 mM sodium acetate, 100 mM NaCl, 10% glycerol, 0.02% NaN$_3$, 0.2% Triton X-100 reduced, pH 5.0). Once the sample had been applied the column was washed with 5 mL buffer A before elution of bound protein a gradient from 0 to 100% buffer B (as A, with 2 M NaCl) over 14 mL. PBP1B containing fractions were pooled, dialysed in 3 mL volume dialysis cassettes (D-Tube maxi, molecular weight cut-off 6–8 kDa; Merck) against storage buffer (20 mM sodium acetate, 500 mM NaCl, 20% glycerol, pH 5.0) and stored at −80 °C.

*LpoA(sol)*: His-LpoA(sol) was purified from strain BL21(DE3) pET28His-LpoA (sol) grown in 1.5 L of LB supplemented with kanamycin at 30 °C to an OD$_{578}$ of 0.4–0.6. Overproduction was induced by adding 1 mM of IPTG and incubating the cells for 3 h at 30 °C. Cells were harvested and resuspended in buffer I (25 mM Tris/HCl, 10 mM MgCl$_2$, 500 mM NaCl, 20 mM imidazole, 10% glycerol, pH 7.5). DNase, protease inhibitor cocktail (Sigma) and PMSF was added before cells were disrupted by sonication. The lysate was centrifuged (130,000 × *g*, 60 min, 4 °C) and

the supernatant was applied to a 5 mL HisTrap HP column (GE Healthcare) attached to an ÄKTA PrimePlus (GE Healthcare) at 1 mL min$^{-1}$. The column was washed with buffer I before stepwise elution of bound proteins with buffer II (25 mM Tris/HCl, 10 mM MgCl$_2$, 500 mM NaCl, 400 mM imidazole, 10% glycerol, pH 7.5). Fractions containing the protein were pooled and dialysed against IEX buffer A (20 mM Tris/HCl, pH 8.0), supplemented with 2 U mL$^{-1}$ of thrombin (Novagen) to remove the His-tag, and applied to a 5 mL HiTrap Q HP column (GE Healthcare). The column was washed with 85% IEX buffer A and 15% IEX buffer B (20 mM Tris/HCl, 500 mM NaCl, pH 8.0) before a linear gradient from 15 to 100% B over 150 mL was applied. The eluted LpoA protein was pooled and concentrated for application to a Superdex200 HiLoad 16/600 column at 1 mL min$^{-1}$ for size exclusion chromatography in a buffer containing 25 mM Tris/HCl, 10 mM MgCl$_2$, 500 mM NaCl, 10% glycerol at pH 7.5. Finally, the protein was dialysed against storage buffer (25 mM HEPES/NaOH, 200 mM NaCl, 10% glycerol at pH 7.5) and stored at −80 °C.

*LpoB(sol)*: His-LpoB(sol) was purified from strain BL21(DE3) pET28His-LpoB (sol) following a similar protocol as described for LpoA(sol), with some modifications. After the first purification step with a HisTrap HP column, fractions containing the protein were pooled, supplemented with 2 U mL$^{-1}$ of thrombin (Novagen) to remove the His-tag, dialysed against 25 mM Tris/HCl, 100 mM NaCl, pH 8.3 and applied to a 5 mL HiTrap Q HP column (GE Healthcare) attached to an ÄKTA Prime (GE Healthcare) at 0.5 mL min$^{-1}$. LpoB was collected in the flow-through, concentrated for application to a Superdex200 HiLoad 16/600 column at 1 mL min$^{-1}$ for size exclusion chromatography in a buffer containing 25 mM HEPES/NaOH, 1 M NaCl, 10% glycerol at pH 7.5. Finally, the protein was dialysed against storage buffer (25 mM HEPES/NaOH, 200 mM NaCl, 10% glycerol at pH 7.5) and stored at −80 °C.

*His-PBP3:* Cells of XL1-Blue + pMvR-1 were grown in 5 L of LB medium with chloramphenicol and 5% glycerol at 30 °C to an OD$_{578}$ of 0.6–0.8. His-PBP3 was overproduced by adding 0.05 mM of IPTG and incubating the cells overnight at 30 °C. Cells were harvested by centrifugation (10,000 × *g*, 15 min, 4 °C) and the pellet was resuspended in 80 mL buffer I (25 mM HEPES/NaOH, pH 8.0) and centrifuge as before. The pellet was resuspended in 80 mL buffer II (25 mM HEPES/NaOH, 1 M NaCl, pH 8.0) and a small amount of DNase, protease inhibitor cocktail (Sigma, 1/1000 dilution) and 100 μM PMSF was added before cells were disrupted by sonication (Branson Digital). The lysate was centrifuged (130,000 × *g*, 1 h, 4 °C) and the resulting membrane pellet was resuspended in extraction buffer (25 mM HEPES/NaOH, 10 mM MgCl$_2$, 1 M NaCl, 2% Triton X-100, pH 8.0) with protease inhibitor cocktail and incubated for 5 h at 4 °C with mixing. The extract was centrifuged (130,000 × *g*, 1 h, 4 °C) and the resulting supernatant was applied to 3 mL of Ni-NTA superflow beads (QIAGEN), supplemented with 20 mM imidazole and protease inhibitor cocktail and incubated for 18 h at 4 °C. Beads were washed with 3 × 10 mL wash buffer I (25 mM HEPES/NaOH, 1 M NaCl, 10 mM MgCl$_2$, 20 mM imidazole, 10% glycerol, 0.2% Triton X-100 reduced, pH 8.0) and 3 × 10 mL wash buffer II (wash buffer I but with 40 mM imidazole). Bound protein was eluted with 10 × 3 mL elution buffer (wash buffer I but with 400 mM imidazole) into tubes containing 1 mM EGTA. Appropriate fractions were pooled and dialysed into storage buffer (25 mM HEPES/NaOH, 1 M NaCl, 10 mM MgCl$_2$, 10% glycerol, pH 8.0) and stored in aliquots at −80 °C.

*FtsN-His:* Cells of BL21(DE3) pFE42 were grown in 2 L of LB medium with ampicillin at 37 °C to an OD$_{578}$ of 0.4. FtsN-His was overproduced by adding 1 mM IPTG to the cell culture followed by a further incubation for 2 h at 37 °C. Cells were harvested by centrifugation (10,000 × *g*, 15 min, 4 °C) and the pellet was resuspended in buffer I (25 mM Tris/HCl, 1 M NaCl, pH 6.0). A small amount of DNase, protease inhibitor cocktail (Sigma, 1/1000 dilution) and 100 μM PMSF was added before cells were disrupted by sonication (Branson digital). The lysate was centrifuged (130,000 × *g*, 1 h, 4 °C). The resulting membrane pellet was resuspended in extraction buffer (25 mM Tris/HCl, 1 M NaCl, 40 mM imidazole, 1% Triton X-100, pH 6.0) and incubated overnight with mixing at 4 °C. The sample was centrifuged (130,000 × *g*, 1 h, 4 °C) and the supernatant applied to a 5 mL HiTrap HP column (GE Healthcare, USA) attached to an ÄKTA Prime (GE Healthcare), at 1 mL min$^{-1}$. The column was washed with four volumes of extraction buffer, followed by four volumes of wash buffer I (25 mM Tris/HCl, 1 M NaCl, 40 mM imidazole, 0.25% Triton X-100, pH 6.0). Bound protein was eluted stepwise with elution buffer (25 mM Tris/HCl, 1 M NaCl, 400 mM imidazole, 0.25% Triton X-100, pH 6.0). FtsN-His was dialysed into storage buffer (25 mM HEPES/NaOH, 500 mM NaCl, 10% glycerol, pH 7.5) and stored in aliquots at −80 °C.

*FtsN-His(sol):* Cells of BL21-A1 pHis17-ECN2 were grown in 2 L of LB medium supplemented with ampicillin at 30 °C to an OD$_{578}$ of 0.5. FtsNΔ1-57-His was overproduced by adding 0.2% arabinose to the cell culture followed by a further incubation for 3 h at 30 °C. Cells were harvested by centrifugation (10,000 × *g*, 15 min, 4 °C) and the pellet was resuspended in buffer I (25 mM Tris/HCl, 500 mM NaCl, pH 6.0). A small amount of DNase, protease inhibitor cocktail (Sigma, 1/1000 dilution) and 100 μM PMSF was added before cells were disrupted by sonication (Branson digital). The lysate was centrifuged (130,000 × *g*, 1 h, 4 °C). The resulting supernatant was applied to 1.5 mL of Ni-NTA superflow beads (Qiagen), supplemented with 10 mM imidazole and incubated for 18 h at 4 °C. Beads were washed with 7 × 10 mL wash buffer (25 mM Tris/HCl, 500 mM NaCl, 20 mM imidazole, pH 6.0) and bound protein eluted with 10 × 1 mL elution buffer (25 mM Tris/HCl, 500 mM NaCl, 300 mM imidazole, pH 6.0). Appropriate

fractions were pooled and dialysed into storage buffer (25 mM HEPES/NaOH, 500 mM NaCl, 10% glycerol, pH 6.0) and stored in aliquots at −80 °C.

*His-ZipA* and *WALP-ZipA-His*: BL21(DE3)pLysS cells containing plasmid pET15Zip[66] were grown in 1 L of autoinduction medium (Miller LB medium supplemented with 0.5% glycerol, 0.05% glucose and 0.2% α-lactose) supplemented with ampicillin for 18 h at 30 °C. Cells were harvested by centrifugation (6200 × g, 15 min, 4 °C) and the pellet was resuspended in buffer I (50 mM Tris/HCl, 20% sucrose, pH 8.0). After addition of 200 μM PMSF, 1/1000 dilution of protease inhibitor cocktail (Sigma-Aldrich, St. Louis, MO) and DNase, the cells were disrupted by sonication (Branson digital). The cell lysate was centrifuged (130,000 × g, 60 min, 4 °C) and the supernatant was discarded. The membrane pellet was resuspended in extraction buffer (50 mM Tris/HCl, 150 mM NaCl, 2% Triton X-100 reduced (Sigma-Aldrich, St. Louis, MO), 10% glycerol, pH 8.0) and incubated overnight with mixing at 4 °C. Resuspended sample was centrifuged (130,000 × g, 60 min, 4 °C) and the supernatant was incubated with 5 mL of Ni-NTA Superflow (Qiagen) for 2 h at 4 °C with gentle stirring, which had been pre-equilibrated in extraction buffer. The resin was poured into a gravity column and washed with 10 volumes of wash buffer (50 mM Tris/HCl, 150 mM NaCl, 0.05% Triton X-100 reduced, 10% glycerol, 50 mM imidazole, pH 8.0). Bound protein was eluted with elution buffer (50 mM Tris/HCl, 150 mM NaCl, 0.05% Triton X-100 reduced, 10% glycerol, 600 mM imidazole, pH 8.0). Eluted protein was dialysed into storage buffer (25 mM HEPES/NaOH, 150 mM NaCl, 10% glycerol, pH 8.0), concentrated four-fold using a VivaSpin-6 column (MWCO 6000 Da) and stored at −80 °C. WALP-ZipA-His was purified from *E. coli* LOBSTR cells transformed with plasmid pPZW22 following a similar procedure. WALP-ZipA was obtained by overnight digestion of WALP-ZipA-His with thrombin at 20 °C.

*ZipA, sZipA and sZipA-His*: BL21(DE3) cells containing plasmid pPZW05 or pPZW06 were grown in 2 L of LB medium supplemented with kanamycin at 37 °C to an OD$_{578}$ of 0.4–0.5. Protein overproduction was induced by addition of 0.5 mM IPTG to the cell culture which was further incubated for 2 h at 37 °C. For ZipA, the procedure was similar as described for His-ZipA except that 2 U mL$^{−1}$ of restriction grade thrombin (Merck Millipore, Darmstadt, Germany) were added to the Ni-NTA eluted protein to remove the oligohistidine tag during dialysis against 3 L of dialysis buffer I (50 mM Tris/HCl, 150 mM NaCl, 10% glycerol, 10 mM EGTA, pH 8) for 20 h at 4 °C. Sample was dialysed against 3 L of dialysis buffer II (10 mM sodium acetate, 150 mM NaCl, 10% glycerol, pH 4.8) for 20 h at 4 °C and applied in buffer A (10 mM sodium acetate, 150 mM NaCl, 10% glycerol, 0.05% Triton X-100 reduced, pH 4.8) to a 5 mL HiTrap SP HP column using an ÄKTA Prime (GE Healthcare Bio-Sciences) for cation exchange chromatography (flow rate 1 mL min$^{−1}$). The protein eluted in a gradient from 150 mM to 2 M NaCl. Protein-containing fractions were dialysed against storage buffer (25 mM HEPES/NaOH, 150 mM NaCl, 10% glycerol, pH 8.0), concentrated four-fold using a VivaSpin-6 column (MWCO 6000 Da) and stored at −80 °C. sZipA was purified from the supernatant of the cell lysate centrifugation using the protocol for ZipA but omitting detergent in buffers. sZipA-His was purified following a similar procedure but no thrombin was added to the sample during dialysis against dialysis buffer I.

### In vivo and in vitro interaction assays

In vivo cross-linking/co-immunoprecipitation assays were performed using dithiobis(succinimidyl) propionate (DSP; ThermoFischer Scientific) as cross-linker reagent. DSP is able to diffuse across the cell membrane allowing for cross-linking reactions in the cytoplasm[67]. *E. coli* BW25113 cells were grown in 150 mL of Lennox LB medium (Fischer Scientific) at 37 °C to an OD of 0.5. Cells were harvested by centrifugation (4000 × g, 20 min, 4 °C) and resuspended in 6 mL cold CL buffer I (50 mM NaH$_2$PO$_4$, 20% sucrose, pH 7.4). Freshly prepared DSP solution (10 mg mL$^{−1}$ in DMSO) was added and cells were incubated for 1 h with mixing. Cross-linked cells were harvested by centrifugation (4000 × g, 15 min, 4 °C) and resuspended in 6 mL of CL buffer II (100 mM Tris/HCl, 10 mM MgCl$_2$, 1 M NaCl, pH 7.5). Protease inhibitor cocktail (Sigma), a small amount of DNase and 100 μM PMSF were added. The cells were disrupted by sonication and membranes were ultracentrifuged (90,000 × g, 60 min, 4 °C) and resuspended in 2.5 mL of CL buffer III (25 mM Tris/HCl, 10 mM MgCl$_2$, 1 M NaCl, 20% glycerol, 1% Triton X-100, pH 7.5). Membranes were extracted overnight at 4 °C with mixing. After a second centrifugation step (90,000 × g, 60 min, 4 °C) the supernatant was taken and diluted with CL buffer IV (75 mM Tris/HCl, 10 mM MgCl$_2$, 1 M NaCl, pH 7.5). The specific antibodies were added and the sample was incubated for 5 h at 4 °C. As a control, sample was incubated without antibody. One hundred microliters of protein G-coupled agarose beads, previously washed with cold water and CL wash buffer (a 2:1 mix of CL buffer III and IV), were added to the membrane fraction and the sample was incubated overnight at 4 °C with mixing. The beads were centrifuged and the supernatant sample was collected. The beads were then washed with 10 mL of CL wash buffer and boiled for 10 min in 50 μL of sample buffer for SDS-PAGE. The supernatant was collected after centrifugation (9,600 × g, 5 min, room temperature) and analysed by SDS-PAGE followed by western blot and immunodetection. MVC1 antibody (1:5,000) and α-FtsN (1:5,000) were used to detect ZipA and FtsN, and anti-rabbit IgG-HRP TrueBlot (Rockland, 18-8816-33) (1:5,000) was used to detect the antibodies of ZipA and FtsN. Uncropped scans of the western blots shown in Fig. 1 are supplied as Supplementary Fig. 9.

For in vitro pulldown assays proteins were mixed at a final concentration of 1 μM in 200 μl of binding buffer (10 mM HEPES/NaOH, 10 mM MgCl$_2$, 150 mM NaCl, 0.05% or 1% Triton X-100, pH 7.5). Samples were incubated at room temperature for 10 min before addition of cross-linker, 0.2% (w/v) formaldehyde (Millipore Sigma), followed by incubation at 37 °C for 15 min. Excess cross-linking was blocked by addition of 100 mM Tris/HCl, pH 7.5. Samples were applied to 100 μL of washed and equilibrated Ni-NTA superflow beads (Qiagen, Hilden, Germany), and incubated overnight at 4 °C with mixing. Beads were washed eight times with 1.5 mL of wash buffer (10 mM HEPES/NaOH, 10 mM MgCl$_2$, 150 mM NaCl, 0.05% or 1% Triton X-100, 50 mM imidazole, pH 7.5). Retained proteins were eluted by boiling the beads in SDS-PAGE loading buffer; beads were then removed, and samples resolved by SDS-PAGE. Uncropped pictures of the gels shown in Fig. 1 are supplied as Supplementary Fig. 10.

### PG synthesis assays

[14C]GlcNAc-labelled lipid II[39], dansylated lipid II[68] and ATTO$^{550}$ lipid II[21,38] were prepared as previously published. Continuous fluorescence GTase assays was performed as described[8], using 0.5 μM of each protein (except His-PBP3 that was at 1.5 μM) in buffer with a final concentration of 50 mM HEPES/NaOH pH 7.5, 150 mM NaCl, 25 mM MgCl$_2$ and 0.04% Triton X-100. Briefly, dansylated lipid II was added to start the reactions and the decrease in fluorescence at 37 °C was measured over time using a plate reader (excitation wave length of 330 nm, emission of 520 nm). Endpoint GTase-TPase activity assay was performed as described[69] using 0.75 μM of each protein and a final concentration of 10 mM HEPES/NaOH pH 7.5, 174 mM NaCl and 0.07% Triton X-100 in the reaction buffer. Briefly, 1.2 nmol (11,000 dpm) of [14C]GlcNAc-labelled lipid II were dried in a glass vial using a vacuum concentrator and resuspended in 5 μL of 0.2% Triton X-100. To start the reactions the assayed proteins were added to the resuspended lipid II and further incubated for 60 min at 37 °C with shaking (800 rpm). Reactions were stopped by boiling for 5 min, and further cellosyl digestion, reduction and analysis by HPLC were performed as described in ref. [69]. The following protein concentrations were used in the assays with low concentrations of PBP1A or PBP1B: 0.075 μM PBP1A, 0.038 μM PBP1B, 0.38 μM LpoB, 3.75 μM BSA. In samples with low PG synthase activity (with abundant unused lipid II) the total radioactivity eluted from the HPLC column (C18) differs between samples due to differences in peak 1, the phosphorylated disaccharide pentapeptide. Peak 1 is generated by acid hydrolysis of unused lipid II (or glycan chains ends carrying the C55-PP moiety) after the GTase-TPase reaction because lipid II (without hydrolysis) does not elute from the C18 HPLC column used to separate the muropeptides. In samples with abundant unused lipid II, peak 1 varies due to differences in the efficiency of the acid hydrolysis of lipid II between samples. This effect does not impair the quantification of other peaks (PG products). Tris-Tricine SDS-PAGE was used to separate glycan chains[70], using the same protein concentrations and reaction conditions than in the TPase activity experiment at low PBP1A/PBP1B concentration but in the presence of 1 mM ampicillin to inhibit the TPase activity.

### Cell parameter measurements and protein immunolocalisation

*E. coli* strains BW25113 (wt), BW25113 ΔmrcA and BW25115 ΔmrcB were used for in vivo studies. Wild-type cells were grown in Luria-Bertani (LB) broth, kanamycin was added to the LB to grow mutants. Cells were cultured (overnight, 37 °C) and then diluted (1:75) in fresh pre-warmed medium. Optical density at 600 nm (OD$_{600}$) was measured at intervals with a CO8000 Cell Density Meter (WPA biowave). To attain exponential balanced growth the cultures were grown in a shaking water bath with aeration and maintained at OD$_{600}$ values below 0.3 by suitable dilutions with pre-warmed medium for at least 4 mass doublings. At this time the cultures were divided in two portions, and maintained at exponential balanced growth phase for 120 min with or without aztreonam (0.3 μg mL$^{−1}$) depending on the strain. Samples were removed from cultures at 20 min intervals along 120 min. Samples were removed from cultures at 20 min intervals along 120 min and fixed in 0.75% formaldehyde. The number of particles per volume was determined using a Beckman Coulter Multisizer 3 multichannel analyzer equipped with a 30 μm diameter orifice. Cell samples for immunofluorescence microscopy were obtained and processed as described[71]. They were fixed with methanol/acetic acid (4:1) and adhered to poly-L-lysine pre-treated coverslips and permeabilize with lysozyme (100 μg mL$^{−1}$, 2 min). Non-specific binding sites were first blocked by incubating cells in 2% bovine albumin (BSA, Serva) in PBS 1 × (20 min) followed by incubation (overnight, 4 °C) with purified antibodies diluted in blocking solution. PBP1A, PBP1B, FtsN, FtsZ, ZipA and FtsA proteins were detected with antibodies α-PBP1A (diluted 1:400), α-PBP1B (1:400), MVG1 (1:500), MVC2 (1:400), MVC1 (1:500) and MVC3 (1:200), respectively. Unbound primary antibodies were removed by extensive washing, followed by incubation with secondary antibody Alexa 594-conjugate anti-rabbit (Invitrogen A-11037) to detect the proteins (red signal). Coverslips were then mounted in Vectashield medium (Vector Laboratories) and sealed. Cells were imaged with a Hamamatsu 3CCD Digital Camera C7780 coupled to a BX61 Olympus fluorescence microscopy, equipped with a 100 × immersion oil lens. The filter used to detect the red signal (protein) was U-MWTY2. The images were captured and deconvolved with SimplePCI imaging software. Intensity levels and image overlay were adjusted using Adobe Photoshop CS3. For quantification purposes, a fluorescent ring was defined either as a sharp bands that crosses the cell from side to side (close ring) or as two bright dots at both sides of the cell (open ring)[72,73].

### Purification of polyclonal antisera

Affinity-purification of polyclonal antisera α-PBP1A, α-PBP1B, MVG1, MVC2 and MVC1 was performed by a modified

published procedure[74]. Purified proteins were separated by SDS-PAGE and transferred to Immobilon-P membranes. The protein band of interest was visualised by staining with 0.1% Ponceau S and cut out. The membrane strip was blocked with 5% dry milk in PBS. The antiserum (200 µL) was layered on top of the membrane strip and incubated with shaking for 2–3 h. After removal of the antiserum and three washing steps for 10 min with PBS, the antibody was eluted with 200 µL of elution buffer (0.2 M glycine, 1 mM EGTA, pH 2.7) for 10 min. The eluted sample was quickly neutralised with an equal volume of 100 mM Tris base. To increase the specificity, the α-PBP1A and α-PBP1B antibodies were repurified using total extracts of mutants BW25113ΔmrcA and BW25115ΔmrcB, respectively, to remove undesired species. Purified antibodies were stored at −20 °C.

**Spot plate assay of thermosensitive strains.** Cells were grown overnight at permissive temperature (30 °C), the optical density was normalised for each strain assayed in the plate and the cells were spotted in a 10-fold dilution series on Lennox LB plates, which were incubated overnight at permissive or non-permissive temperature (42 °C).

**Detection of preseptal PG synthesis by labelling using HADA.** Cells were streaked on Lennox LB agar plates. Single colonies were used to inoculate 20 mL of media and cultures were incubated overnight at 30 °C. Overnight cultures were diluted 1:500 and grown for 1 h at 30 °C. Depletion of ZipA and/or FtsN was achieved by incubating the cultures at 42 °C for 1.5 h. During this time the cultures were kept in exponential phase ($OD_{578}$ between 0.2–0.4). Samples were diluted to $OD_{578}$ of 0.2 in a final volume of 500 µL, and further incubated with 250 µM of the fluorescent derivative of D-Ala (HADA) for 30 min at 42 °C (long pulse labelling). To remove the excess of HADA, 2 mL of pre-warmed media supplemented with aztreonam (1 µg mL$^{-1}$) were added to the samples, followed by centrifugation (1,900 × $g$, 3 min, 40 °C). The cell pellets were washed a second time and the cells were resuspended in 1 mL pre-warmed media containing aztreonam, and incubated 40 min at 42 °C with shaking. One hundred microliters of a 10 × sodium citrate buffer (805 mM citric acid, 1.19 M NaCl, 410 mM NaOH, pH 2.25) were added to each 1 mL sample. The samples were put immediately on ice and centrifuged (16,200 × $g$, 2 min, 4 °C). Afterwards the samples were washed once with 1.5 mL of sodium citrate buffer (80.5 mM citric acid, 119.4 mM NaCl, 41 mM NaOH, pH 3.0) and twice with PBS (1.7 mM $KH_2PO_4$, 5 mM $Na_2HPO_4$, 150 mM NaCl, pH 7.4). The final cell pellets were resuspended in 12.5 µL PBS and 12.5 µL of 3% paraformaldehyde (diluted in PBS) were added for cell fixation. The cell samples were analysed using a Nikon Eclipse Ti microscope (Nikon Plan Fluor ×100/1.30 Oil Ph3 DLL objective) coupled to a photometrics/Cool SNAP HQ$^2$ CCD camera using the phase contrast and the DAPI channel (DAPI filter set: Chroma 49000, excitation at 350/50 nm, emission 460/50 nm). To quantify the number of preseptal sites per cell length unit (µm), cell lengths were measured and the preseptal PG synthesis zones were counted using the ObjectJ plug-in [https://sils.fnwi.uva.nl/bcb/objectj/] (University of Amsterdam) of the ImageJ software[75]. Preseptal PG synthesis sites were defined as non-fluorescent bands across the whole width of the cell in the fluorescence image and no visible constriction in the phase contrast image. All quantified cells showed growth during the chase period, indicated by brighter cell poles compared to the side wall due to PG incorporation during elongation.

## Data availability
The relevant data supporting the findings of this study are available from the corresponding author.

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

## Acknowledgements

We thank Orietta Massida (University of Trento) and Martin Loose (IST Vienna) for critical reading of the manuscript, Michael VanNieuwenhze's lab for the synthesis of HADA, Alexander Egan and Víctor Hernández-Rocamora (Newcastle University) for purified proteins, and William Margolin (University of Texas) for strains and helpful discussions. This work was supported by the Wellcome Trust (101824/Z/13/Z), the MRC (MR/N002679/1), European Commission (Marie Curie Actions, FP7-PEOPLE-2013-IEF-623868) and NIH (GM 111537).

## Author contributions

M.P and W.V. designed the work. M.P., K.P., M.C and P.P performed experiments. M.P, K.P., M.C., M.V. and W.V. analysed the data. E.B. and M.VN. provided research tools. All authors contributed to writing the manuscript.

## Additional information

**Competing interests:** The authors declare no competing interests.

