## [Peer Review File · Nature Communications]

Reviewers' comments:

Reviewer #1 (Remarks to the Author):

In this manuscript, Pazos et al. seek to elucidate molecular details underlying pre-septal cell wall synthesis in *E. coli*. To this end, they focus on characterizing ZipA and FtsA-FtsN as partially redundant membrane anchors that bridge interactions between the Z-ring and the bifunctional peptidoglycan synthases responsible for pre-septal PG synthesis. The authors use *in vivo* and *in vitro* immunoprecipitation techniques to demonstrate interaction of ZipA with PBP1A, PBP1B, and PBP3 and immunofluorescence microscopy to demonstrate localization of membrane anchors and PG synthases to likely sites of pre-septal PG synthesis. They find moderate effects of ZipA on the GTase activities of PBP1A/B, which are additive when combined with FtsN, using *in vitro* PG synthesis assays. Finally, the authors assess functional redundancy of ZipA and FtsN in directing pre-septal PG synthesis using *in vivo* genetic experiments in a FtsA* background that renders zipA or ftsN individually non-essential. The authors conclude that ZipA and FtsN drive pre-septal synthesis at the Z-ring through redundant activation of bifunctional PBPs.

This paper is quite clearly and logically written and the final model figure does a fantastic job of summarizing the authors' thinking on how their biochemical, genetic, and cell biological data fit together. The combination of *in vivo* analyses with *in vitro* PG synthesis assays is a welcome pairing towards answering mechanistic questions of pre-septal cell wall synthase activation and the authors provide some novel details about interactions between ZipA and PBPs. The work will certainly be of interest to those in the field, but may be of limited interest to the broader bacteriology community. There are a few areas where the data could be strengthened to better support the authors' conclusions.

Major comments:

1) In figure 1c-d, there is substantial variation in the efficacy of pulling down ZipA. The amount of His-ZipA pull down between c and d are quite different, while sZipA-His does not seem to be efficiently pulled down at all, making it difficult to draw conclusions about the amount of PBP1A or B pulled down. Is there a technical explanation for the variability? The authors use these data to argue that the TM and/or periplasmic regions of ZipA may be important for interaction with PBPs. The authors might use a ZipA chimera with the TM domain of another protein fused to it in the *in vitro* pull downs as bolstering evidence for a specific role for its TM region in the PBP interaction.

2) In the experiments in Figure 2, cells are treated with aztreonam for up to 40 minutes before being processed for IFM using antibodies against PBP1A, PBP1B, FtsN, FtsZ, or ZipA. The authors then report "% of cells showing (pre-)septal rings containing" the various proteins. However, it is not clear how the authors distinguished "(pre-)septal rings" from other foci that may be near midcell. In the key cases of PBP1A and PBP1B, while the number of foci per cell appear to be reduced upon treatment with aztreonam, there is no previously characterized marker of pre-septal sites in these cells to demonstrate that the PBP1A/B foci are at sites of pre-septal cell wall synthesis. Ideally, these experiments should be repeated with a pre-septal marker, such as FtsZ, in the same cells to convincingly demonstrate pre-septal localization. Alternatively, or ideally in addition, demographs might be helpful for presenting this data across the population and demonstrating a regular pattern of localization similar to that observed for FtsZ.

3) In the *in vitro* assays in Figures 3 and 4, there are key controls missing. Specifically, all of the reactions contain PBP1A or PBP1B and the negative controls of ZipA, FtsN, or PBP3 alone (or in the relevant combinations with each other) are not included. The authors conclude that ZipA and FtsN have mild to moderate ability to activate PBP1A and/or B. However, without the negative controls, it is possible that this apparent enhancement of PBP1A/B activity is due to a contaminant in the ZipA or FtsN preps. Additionally, Figure 3 has the data presented as normalized activity. Why not present the raw activity (ideally in a scatter plot including all data points, not a bar graph) so the

reader can ascertain more directly the effects of possible activators?

4) The authors imply in the discussion and their model that pre-septal cell wall synthesis is essential for *E. coli* viability. That is, the reason zipA and ftsN are synthetic lethal in the ftsA* background is that one or the other must be present to direct pre-septal cell wall synthesis. Can the authors expand the discussion a bit to flesh out why pre-septal cell wall would be essential for viability and/or include discussion of relevant prior data supporting that hypothesis?

Minor comments:

1) Lines 265-266, the authors say that "ZipA and FtsN show similar phylogenetic distribution being present mainly in the gammaproteobacteria". This does not seem accurate. ZipA is found only in Enterobacteria, while Möll and Thanbichler (2009 *Mol Micro*) identified FtsN homologs throughout the gamma, alpha, beta, and delta-proteobacteria. Please revise or eliminate this argument.

2) In the experiments for Figure 1, the labels need to be clarified. What does "App." stand for in c-e? For "SN" in 1b, is that the unbound proteins or input?

3) In the in vitro IPs in figure 1-c, cross-linking was used. Have you attempted these pulldowns without cross-linking?

4) In figure 5, it would be easier for the reader if the abbreviated genotype were used in the label (e.g. ftsAE124A instead of WM2935), with both the strain number and genotype listed in the legend.

5) Line 201 – Add "for pre-septal cell wall synthesis" so that the sentence reads – "... could be essential for pre-septal cell wall synthesis in the absence of zipA.

6) Line 212, line 387, and Figure 5 – How is the quantification done to determine pre-septal PG synthesis zones? It would be easier to see bands of pre-septal PG synthesis if they were positively stained (i.e. instead of pre-labeling cells and looking for clearing of label, pulse label with HADA after the shift to restrictive temperature). In the double mutant (g) there is somewhat patchy HADA distribution, making it difficult to clearly tell if pre-septal zones are eliminated completely.

7) In the methods, it is preferable to describe each method at least in brief, rather than only referring to prior publications so that the reader does not have to follow a paper trail to find the relevant details.

Reviewer #2 (Remarks to the Author):

An early step in *E. coli* cell division is synthesis of a ring of "pre-septal" peptidoglycan at the nascent division site. Previous studies have shown synthesis of pre-septal PG requires the FtsZ ring, ZipA and a bifunctional penicillin-binding protein, either PBP1A or PBP1B. Both of these PBPs are bifunctional enzymes with transglycosylase activity for polymerizing glycan strands and transpeptidase activity for crosslinking stem peptides. Both PBP1A and PBP1B also contribute to elongation. The manuscript under review addresses two questions: How are PBP1A and PBP1B recruited to the division site? How is their PG synthesis activity turned on once they get there?

Pazos et al. present data consistent with a model in which both PBPs are recruited and activated by binding directly to ZipA. Multiple lines of evidence support this conclusion. (a) Both PBPs co-localize with ZipA to division sites when the monofunctional transpeptidase PBP3 (also called FtsI) is inactivated with the beta-lactam aztreonam. (b) ZipA binds to PBP1A and PBP1B in multiple

assay formats. (c) Purified ZipA stimulates PG synthesis by PBP1A and PBP1B in vitro in some assay formats, but not in others. Additional lines of evidence argue FtsN can substitute for ZipA in recruiting and activating pre-septal PG synthesis by PBP1A and PBP1B, so there is functional redundancy of ZipA and FtsN in promoting synthesis of pre-septal PG.

Overall, this study is an important advance towards understanding an enigmatic phase of the division process. The redundancy of approaches spanning from binding assays, to enzyme assays and cytological assays are a strength. Nevertheless, some of the conclusions do not seem fully justified to me, and some points need further clarification.

Major concerns

1. Line 28-29 of the Abstract: "Here we show that PBP1A and PBP1B interact with ZipA and localize to pre-septal sites in cells with inhibited PBP3." The relevant experiment is in Figure 2, which does indeed show both PBPs localize to division sites when PBP3 is inhibited. Importantly, however, no experiments are included to show this recruitment requires ZipA. Elsewhere, for example in Figure 1, the authors show the PBPs interact with ZipA. So it is certainly logical for them to make the connection that this interaction recruits the PBPs. But whether this is indeed the case is not yet known. I recommend doing localization in a zipA(Ts) mutant or depletion strain. The experiment in Figure 5 demonstrates ZipA is needed for PIPS, but does not show whether this is because PBP1A/1B fail to localize or localize but remain inactive.

2. In a previous paper, the senior author reported that recruitment of PBP1B to the division site requires PBP3 (Bertsche et al., Mol Microbiol 2006). This was an important enough finding to warrant mention in the abstract of that paper: "...localization of PBP1B at the septation site depended on the physical presence of PBP3, but not on the activity of PBP3." The new manuscript under review conflicts with the earlier study. This discrepancy needs to be addressed.

3. I have several concerns about how to interpret experiments showing that purified ZipA and FtsN stimulate PG synthesis by PBP1A and PBP1B in vitro. The relevant data are in Figures 3 and 4. First, it seems to me that the stimulation is very small, at best 2.6 fold (1A + ZipA in Figure 2A) but sometimes much less, such as only 1.4 fold (1A + FtsN in Figure 2A). Moreover, this stimulation is only seen in one assay format (Figure 2) but not in two other assay formats (Figure 3). The authors have an explanation related to dimerization of the PBPs, and this might be true. But then again, it might not. This raises the question: Is a weak stimulation that is only observed under limited conditions convincing? The authors would probably say yes because BSA does not stimulate. I consider BSA a weak control. Far better would be to use mutant forms of PBPs, ZipA and FtsN to determine whether the stimulation depends on logical domains. For example, soluble ZipA should not stimulate either PBP, nor should full-length ZipA stimulate truncated PBPs that are lacking their membrane anchor domains.

4. Figure 3b. What do the authors make of their finding that ZipA prevents PBP3 from activating PBP1B? Since both proteins would be present in pre-septal sites in vivo, doesn't this finding contradict their central conclusion? In other words, the data seem to imply ZipA would only activate PBP1B when PBP3 is not present, and vice versa. But in real life both proteins would be present and cancel each other out, wouldn't they? Of course in real life FtsN would be present too, which in the in vitro assay overcomes the PBP3-dependent squelching of ZipA. Overall, taking the data at face value, it appears ZipA and PBP3 independently activate PBP1b, unless they are mixed, in which case they cancel each other out until FtsN comes along is very complex. The authors should comment on how they think these various results are to be interpreted.

5. I have difficulty understanding the chromatograms shown in Figure 4a. My confusion arises from the different sizes of peaks 1 and 2. Why is peak 1 so small in the case of PBP1A alone or PBP1A + ZipA? To an outsider it looks like there was a pipetting error such that hardly any substrate was added to the reaction mixture. In the latter trace (PBP1A + ZipA), peak 2 is as large

as peak 1, although both are very small. Doesn't the increase in peak 2 relative to peak 1 mean that substrate was converted to product? If so, then ZipA stimulates PBP1A, contrary to what the authors conclude. Please clarify how these data should be read.

6. Line 125 refers to Table S3, which quantifies septal localization of PBP1A and 1B under various conditions. It looks to me like Table S3 says PBP1A localizes to division sites in 77% of wild-type cells growing normally. But I think the value should be closer to 0%. Please check this and modify the table or the text to avoid confusion.

Minor points

7. Line 63. The passage is written in a way that implies FtsW is a lipid II flippase but not a transglycosylase, but this is a matter of contention. It would be better to say that FtsW is proposed by some to be a lipid II flippase but by others to be a PG transglycosylase.

8. Line 139. Here the text states that PBP1A and PBP1B are recruited to division sites "...presumably linked to the Z-ring by ZipA, although FtsN may also contribute to their recruitment." As noted above, what about the previous claim that localization of PBP1B required the physical presence of PBP3?

9. Related to the PBP1b-PBP3 interaction, I think the manuscript under review brings to 8 the number of proteins the senior author has reported bind directly to PBP1B: Itself (dimerization), LpoB, MltA, MipA, PBP3, FtsN, FtsW and now ZipA. I think this should be noted somewhere. First, a global picture of PBP1b's reported interaction partners provides important context for thinking about how PBP1b might be recruited and regulated in vivo. Second, the fact that PBP1b has some many binding partners means some assays in the manuscript under review are missing components that a likely to be relevant. This could affect the results of binding and PG synthesis assays. Third, readers will realize that the interaction assays used in this paper to show ZipA binds to PBP1b have a history of finding many such interactions, which could mean PBP1b is part of a large complex or that the assays are prone to false positives. I don't think this manuscript is the place to take up the meaning of such a large number of interaction partners, but I do think the existence of the growing set of interactions should be noted. (Refs for interactions: Laclercq et al., 2017; Typas et al., 2010; Muller et al, 2007; Bertsche et al., 2006; Bertsche et al., 2005; Vollmer et al, 1999)

10. Line 79. Do the authors mean to include RodA as division protein?

11. Line 249. "If zipA is absent, FtsN becomes the only linker between the cytosolic proto-ring (formed by FtsA* and FtsZ) and the PG synthases..." If that's the model, why does Figure 1a show PBP3 bound directly to FtsA? Maybe the passage should specify "bifunctional" PG synthases.

12. Line 145 and Figure 3. ZipA increases GTase activity slightly when added to PBP1A or 1B in vitro. The authors interpret this to mean ZipA binds to and activates both proteins. This is surprising and interesting because the two PBPs do not have homologous membrane anchors. The text should note this. Or do the authors suspect that ZipA interacts directly with the GTase domains? These are at least homologous.

13. Please revise the part of the methods section that deals with strain construction to make it easier for the reader to understand. I had to constantly bounce back and forth between the text and the strain table, which was in a separate document. For example, line 277 now reads: To produce strain MPW22, WM2935 was transformed with pCH32 and transduced with the P1 lysate from the strain WM1304 (Δ zipA::aph). But what are MPW22, pCH32 and MW2935? The sentence would be easy to follow if key genotypes were included in the text, e.g., "To produce strain MPW22 [ftsA E124A zipA::aph/pCH32(zipA, ftsZ)], WM2935 (ftsA E124A) was transformed with pCH32 (zipA, ftsZ) and transduced with the P1 lysate from the strain WM1304 (Δ zipA::aph)."

14. Line 296. Was the entire construct sequenced? I assume only the “insert” was sequenced even though the entire plasmid was made by PCR. Please clarify.

15. The Lpo overproduction plasmids are pET28 derivatives but Table S2 indicates the His tag is at the N-terminus. Is that correct? Please check other plasmids and figures for accuracy because Table S2 lists pET28 derivatives that produce ZipA-His or soluble ZipA-His, but in Figure 1 there is His-ZipA but sZipA-His. There might not be any mistakes here because a variety of overproduction plasmids were used in this study.

Reviewer #3 (Remarks to the Author):

This manuscript investigates a somewhat mysterious process that has been referred to as PIPS, which is an FtsZ dependent, but PBP3 independent, incorporation of DAP at the division site. A similar process has been described in *Caulobacter* but it's too early to say whether it is the same. In this paper, the authors investigate the requirements for this process in *E. coli*. In part they are guided by a paper from Young's lab that shows that PIPS depends upon ZipA but can also occur in FtsA* strains that bypass ZipA. They first detect interactions between PBP1B and ZipA, PBP1A and ZipA as well as PBP3 and ZipA by chemical crosslinking and IP, but not with any of these with FtsN. These interactions are by testing direct interactions using purified proteins and a pull down assay. They then show by one assay that the GTase, but not the TPase, activities of PBP1A and 1B are stimulated in vitro. 1A is stimulated 2X by ZipA and marginally by FtsN whereas 1B is stimulated by FtsN and marginally by ZipA. In another assay (lower protein concentration) 1A is not affected whereas 1B is stimulated by FtsN. They then show that 1A and 1B are localized to the division site in the presence of aztreonam. This is the most dramatic localization of these proteins to potential division sites that I have seen. Also, they fuse HADA labelling to show that PIPS occurs in a ftsA* strain if FtsN or ZipA is present but not if both are absent. They then suggest that PIPS is due to 1A or 1B being recruited to the division site by ZipA or FtsN and being stimulated there.

One of my main concerns with the paper is there is what is not mentioned. First of all there is no mention of FtsW as a potential polymerase. Instead it is dismissed based upon the failure of one lab to observe activity in an in vitro assay. No mention is made of the fact that a close paralog of FtsW, RodA has been shown to be a polymerase by one lab (Meeske et al. 2016) and inferred to be a polymerase by another (Emani et al. 2017). Thus, it is possible that the PIPS seen in the absence of ZipA is due to FtsW (it should be localized). It is also possible that PIPS seen in the absence of FtsN is also due to FtsW. Of course some other protein must do the crosslinking. Second, a paper by the Bernhardt lab shows that proteins from *Pseudomonas*, PBP1B and a protein that substitutes for LpoB, work in *E. coli* and can take the place of PBP1B and LpoB. It is unlikely that the proteins from this organism would retain the specific contacts with the division proteins from *E. coli*.

The crosslinking of ZipA with 1A, 1B and PBP3 is surprising for several reasons. One is very little of ZipA is in the periplasm so the crosslinking must occur in the cytoplasm. The crosslinking reagent is for primary amines and ZipA only has no lysines in the periplasm and none in the transmembrane region.

The authors suggest the transmembrane region is responsible for interactions between ZipA and its partners. It is not clear how this would stimulate the enzyme activity of 1A since the enzymatic domains are not near the membrane.

The localization of 1A and 1B is the most dramatic I've seen. It would be nice to show that this coincides with the Z ring. What is surprising is that 1A displays better localization than 1B, which

has been linked to division in the past.

I worry about experiments that depend so much on biochemistry and have no mutants to support the work to show the physiological significance.

Rebuttal - Reviewer's comments in blue text; our reply in black text

Reviewer #1 (Remarks to the Author):

In this manuscript, Pazos *et al.* seek to elucidate molecular details underlying pre-septal cell wall synthesis in *E. coli*. To this end, they focus on characterizing ZipA and FtsA-FtsN as partially redundant membrane anchors that bridge interactions between the Z-ring and the bifunctional peptidoglycan synthases responsible for pre-septal PG synthesis. The authors use *in vivo* and *in vitro* immunoprecipitation techniques to demonstrate interaction of ZipA with PBP1A, PBP1B, and PBP3 and immunofluorescence microscopy to demonstrate localization of membrane anchors and PG synthases to likely sites of pre-septal PG synthesis. They find moderate effects of ZipA on the GTase activities of PBP1A/B, which are additive when combined with FtsN, using *in vitro* PG synthesis assays. Finally, the authors assess functional redundancy of ZipA and FtsN in directing pre-septal PG synthesis using *in vivo* genetic experiments in a FtsA* background that renders zipA or ftsN individually non-essential.

The authors conclude that ZipA and FtsN drive pre-septal synthesis at the Z-ring through redundant activation of bifunctional PBPs.

This paper is quite clearly and logically written and the final model figure does a fantastic job of summarizing the authors' thinking on how their biochemical, genetic, and cell biological data fit together. The combination of *in vivo* analyses with *in vitro* PG synthesis assays is a welcome pairing towards answering mechanistic questions of pre-septal cell wall synthase activation and the authors provide some novel details about interactions between ZipA and PBPs. The work will certainly be of interest to those in the field, but may be of limited interest to the broader bacteriology community. There are a few areas where the data could be strengthened to better support the authors' conclusions.

Reply: We thank the reviewer for their insightful comments. We performed additional experiments and added these data into the revised manuscript to better support our conclusions.

Major comments:

1) In figure 1c-d, there is substantial variation in the efficacy of pulling down ZipA. The amount of His-ZipA pull down between c and d are quite different, while sZipA-His does not seem to be efficiently pulled down at all, making it difficult to draw conclusions about the amount of PBP1A or B pulled down. Is there a technical explanation for the variability?

Reply: We see the point made by the referee but refrain from drawing quantitative conclusions from this assay. In fact, the amount of His-ZipA is only lower in the elution of the experiment with PBP1A (in panel c), whereas in the other samples (with His-ZipA alone in panels c and d, or with PBP1B in panel d) the differences are small. As pointed out by the referee, we consistently obtained less sZipA-His with C-terminal His-tag in the elution fractions compared to His-ZipA (although the amounts are similar in panels c and d). We do not know the reason for these differences, but they might be due to different strength of binding to Ni-NTA beads between sZipA-His and His-ZipA.

We have now repeated again the experiment with PBP1B and sZipA-His, showing a similar amount of sZipA-His as in other samples, and added this result as new panel d.

The authors use these data to argue that the TM and/or periplasmic regions of ZipA may be important for interaction with PBPs. The authors might use a ZipA chimera with the TM domain of another protein fused to it in the *in vitro* pull downs as bolstering evidence for a specific role for its TM region in the PBP interaction.

Reply: To address this point we cloned and purified a chimera fusion of an artificial transmembrane domain (WALP23) (de Planque *et al.*, 1999, *JBC* 274, 20834-20846) to the cytosolic region of ZipA. We repeated the pulldown assays and show that the WALP23-sZipA chimera did not interact with PBP1A, PBP1B or HisPBP3, supporting a specific role of the transmembrane region of ZipA in interactions with PBPs. These results were added into the revised manuscript (lines 111-117 and Supplementary Fig.1, new panels c and d).

2) In the experiments in Figure 2, cells are treated with aztreonam for up to 40 minutes before being processed for IFM using antibodies against PBP1A, PBP1B, FtsN, FtsZ, or ZipA. The authors then report “% of cells showing (pre-)septal rings containing” the various proteins. However, it is not clear how the authors distinguished “(pre-)septal rings” from other foci that may be near midcell.

Reply: We defined as 'ring' either a sharp band that crosses the cell from side to side (closed ring) or two bright signals (dots) at both sides of the cell (open ring), consistent with the definition of rings used in previous publications (den Blaauwen *et al.* 1999 *J Bacteriol.* 181, 5167-5175; Vicente *et al.* 2006 *J Bacteriol.* 188,19-27). We added a statement to the methods section (lines 456-458) to clarify how we defined the rings, referring to the previous literature. We also added the new Supplementary Fig.4 showing the ring frequency of each immunodetected protein and its relative localisation along the cell length after 40 min of treatment with aztreonam in WT cells. This figure shows that the position of the ring structures from each protein superimposes with the Z-rings at the preseptal positions ($\frac{1}{4}$, $\frac{1}{2}$ and $\frac{3}{4}$ of the cell length). Hence, we conclude that the proteins mentioned by the reviewer do indeed localise at preseptal positions.

In the key cases of PBP1A and PBP1B, while the number of foci per cell appear to be reduced upon treatment with aztreonam, there is no previously characterized marker of pre-septal sites in these cells to demonstrate that the PBP1A/B foci are at sites of pre-septal cell wall synthesis. Ideally, these experiments should be repeated with a pre-septal marker, such as FtsZ, in the same cells to convincingly demonstrate pre-septal localization. Alternatively, or ideally in addition, demographs might be helpful for presenting this data across the population and demonstrating a regular pattern of localization similar to that observed for FtsZ.

Reply: As mention in our reply to the previous point, the quantification of the fluorescence rings along the cell length (Supplementary Fig.4) shows that the relative position of rings containing PBP1A, PBP1B or FtsZ are virtually identical (i.e, at $\frac{1}{4}$, $\frac{1}{2}$ and $\frac{3}{4}$ of the cell length), strongly supporting the localisation of PBP1A and PBP1B at preseptal position.

3) In the *in vitro* assays in Figures 3 and 4, there are key controls missing. Specifically, all of the reactions contain PBP1A or PBP1B and the negative controls of ZipA, FtsN, or PBP3 alone (or in the relevant combinations with each other) are not included. The authors conclude that ZipA and FtsN have mild to moderate ability to activate PBP1A and/or B. However, without the negative controls, it is possible that this apparent enhancement of PBP1A/B activity is due to a contaminant in the ZipA or FtsN preps.

Reply: The reviewer made a valid point. In our assays we compared the activities to samples with PBP1A or PBP1B alone (Figures 3 and 4) and, as the reviewer wrote, we did not include the negative controls (ZipA, FtsN or PBP3 alone). We however would like to note that if these proteins would contain any contamination by an active PG synthases (as the reviewer considered) then this contamination could be expected to enhance PG synthesis in combination with both, PBP1A and PBP1B. However, both PBPs were affected differently by ZipA, FtsN and PBP3 (Figures 3 and 4), hence it is unlikely that these effects on activities were caused by contaminating PG synthases. To rigorously exclude such a possibility, we have performed the control experiments with ZipA, FtsN or

PBP3 alone. We also performed reactions in which we added (i) sFtsN (soluble version of FtsN lacking the TM region) or (ii) sZipA to reactions of PBP1A or PBP1B in the GTase-TPase, SDS-PAGE GTase and fluorescent GTase assays. We added these new data into the revised ms (lines 163-164, 171-173, 191-193, 195-196 and 209-214). They show that (i) sZipA does not enhance the GTase activity of PBP1A or PBP1B (Figure 3), consistent with our finding that the TM region of ZipA is required for interaction with PBPs, (ii) purified ZipA, FtsN and PBP3 do not exhibit PG synthase activity (and, hence, do not contain a contaminating PG synthase) (Supplementary Fig.7b and Supplementary Fig.7c) and (iii) sFtsN has a much weaker stimulatory effect on the GTase-TPase (Supplementary Fig.7a) or GTase (Supplementary Fig.7c) activities of PBP1B, compared to the full length FtsN, consistent with our previously published results (Müller *et al.* 2007, *JBC* 282, 36394-36402).

Additionally, Figure 3 has the data presented as normalized activity. Why not present the raw activity (ideally in a scatter plot including all data points, not a bar graph) so the reader can ascertain more directly the effects of possible activators?

Reply: Showing all raw data would require adding 72 curves, which would expand the Supplemental data file unnecessarily. Instead of showing all curves, we added the curves of the mean fluorescence of 3 independent experiments (new Supplementary Fig.6).

4) The authors imply in the discussion and their model that pre-septal cell wall synthesis is essential for *E. coli* viability. That is, the reason *zipA* and *ftsN* are synthetic lethal in the *ftsA** background is that one or the other must be present to direct pre-septal cell wall synthesis. Can the authors expand the discussion a bit to flesh out why pre-septal cell wall would be essential for viability and/or include discussion of relevant prior data supporting that hypothesis?

Reply: We thank the reviewer for the suggestion. We now discuss in more detail why pre-septal cell wall synthesis would be essential (lines 278-283).

Minor comments:

1) Lines 265-266, the authors say that “ZipA and FtsN show similar phylogenetic distribution being present mainly in the gammaproteobacteria”. This does not seem accurate. ZipA is found only in Enterobacteria, while Möll and Thanbichler (2009 *Mol Micro*) identified FtsN homologs throughout the gamma, alpha, beta, and delta-proteobacteria. Please revise or eliminate this argument.

Reply: We thank the reviewer for this point and as suggested eliminated the argument (lines 305-307).

2) In the experiments for Figure 1, the labels need to be clarified. What does “App.” stand for in c-e? For “SN” in 1b, is that the unbound proteins or input?

Reply: We clarified the labels in the figure legend.

3) In the in vitro IPs in figure1-c, cross-linking was used. Have you attempted these pulldowns without cross-linking?

Reply: We have not performed the assay without cross-linker.

4) In figure 5, it would be easier for the reader if the abbreviated genotype were used in the label (e.g. *ftsAE124A* instead of WM2935), with both the strain number and genotype listed in the legend.

Reply: As suggested we added the genotype to the figure.

5) Line 201 – Add “for pre-septal cell wall synthesis” so that the sentence reads – “... could be

essential for pre-septal cell wall synthesis in the absence of zipA.

Reply: We changed the sentence as suggested (lines 224-225).

6) Line 212, line 387, and Figure 5 – How is the quantification done to determine pre-septal PG synthesis zones?

Reply: We consider a pre-septal PG synthesis zone as a non-fluorescent band across the whole width of the cell in the fluorescence image and no visible constriction in the phase contrast image. We added this information to the ms (lines 499-501).

It would be easier to see bands of pre-septal PG synthesis if they were positively stained (i.e. instead of pre-labeling cells and looking for clearing of label, pulse label with HADA after the shift to restrictive temperature).

Reply: The referee made a good point. We initially tried to label pre-septal bands by a short HADA pulse after the addition of aztreonam. However, for unknown reason we did not observe enhanced labelling at pre-septal sites. Hence, we decided to use the previously published, standard procedure for the detection of pre-septal sites (Potluri *et al.* 2012, *J Bacteriol* 194, 5334-5342 and de Pedro *et al.* 1997, *J Bacteriol* 179, 2823-2834), replacing the previously used D-Cys by HADA and imaging cells after complete PG labelling followed by a chase without label. As expected from these previous publications, the pre-labelling method allowed to visualize pre-septal sites.

In the double mutant (g) there is somewhat patchy HADA distribution, making it difficult to clearly tell if pre-septal zones are eliminated completely.

Reply: There is a gap in the fluorescent signal at the bottom part of this filament but this was not counted as a pre-septal zone because the phase contrast image shows a partial constriction at this site (indicating later stage of septation - presumably cell division was aborted at this site when aztreonam was added). In fact, as shown in panel c, pre-septal zones were not completely eliminated in the double mutant (presumably due to incomplete depletion of *zipA* and *ftsN*), but were significantly reduced.

7) In the methods, it is preferable to describe each method at least in brief, rather than only referring to prior publications so that the reader does not have to follow a paper trail to find the relevant details.

Reply: As suggested, we added more details for the different PG synthesis assays (lines 407-430).

Reviewer #2 (Remarks to the Author):

An early step in *E. coli* cell division is synthesis of a ring of “pre-septal” peptidoglycan at the nascent division site. Previous studies have shown synthesis of pre-septal PG requires the FtsZ ring, ZipA and a bifunctional penicillin-binding protein, either PBP1A or PBP1B. Both of these PBPs are bifunctional enzymes with transglycosylase activity for polymerizing glycan strands and transpeptidase activity for crosslinking stem peptides. Both PBP1A and PBP1B also contribute to elongation. The manuscript under review addresses two questions: How are PBP1A and PBP1B recruited to the division site? How is their PG synthesis activity turned on once they get there?

Pazos *et al.* present data consistent with a model in which both PBPs are recruited and activated by binding directly to ZipA. Multiple lines of evidence support this conclusion. (a) Both PBPs co-localize with ZipA to division sites when the monofunctional transpeptidase PBP3 (also called FtsI) is inactivated with the beta-lactam aztreonam. (b) ZipA binds to PBP1A and PBP1B in multiple assay formats. (c) Purified ZipA stimulates PG synthesis by PBP1A and PBP1B *in vitro* in some assay formats, but not in others. Additional lines of evidence argue FtsN can substitute for ZipA in

recruiting and activating pre-septal PG synthesis by PBP1A and PBP1B, so there is functional redundancy of ZipA and FtsN in promoting synthesis of pre-septal PG.

Overall, this study is an important advance towards understanding an enigmatic phase of the division process. The redundancy of approaches spanning from binding assays, to enzyme assays and cytological assays are a strength. Nevertheless, some of the conclusions do not seem fully justified to me, and some points need further clarification.

Reply: We thank the reviewer for their insightful comments. We hope that the reviewer finds our conclusions justified in the revised manuscript and that the points we make are now clear.

Major concerns

1. Line 28-29 of the Abstract: “Here we show that PBP1A and PBP1B interact with ZipA and localize to preseptal sites in cells with inhibited PBP3.” The relevant experiment is in Figure 2, which does indeed show both PBPs localize to division sites when PBP3 is inhibited. Importantly, however, no experiments are included to show this recruitment requires ZipA. Elsewhere, for example in Figure 1, the authors show the PBPs interact with ZipA. So it is certainly logical for them to make the connection that this interaction recruits the PBPs. But whether this is indeed the case is not yet known. I recommend doing localization in a zipA(Ts) mutant or depletion strain. The experiment in Figure 5 demonstrates ZipA is needed for PIPS, but does not show whether this is because PBP1A/1B fail to localize or localize but remain inactive.

Reply: We thank the reviewer for this excellent point. We performed new experiments to localise PBP1A, PBP1B, ZipA, FtsA and FtsZ in ZipA-depleted cells (the zipA depletion is incomplete with this system as published in: Hale and de Boer 1999, *J Bacteriol* 181, 167-176). PBP1A, PBP1B and residual ZipA localised along the cell length instead of the preseptal positions (where FtsZ and FtsA did localise), suggesting that the absence of preseptal PG synthesis in ZipA-depleted cells is likely due to mislocalisation of the PBPs rather than a lack of activity. We included these data into the ms as new Supplementary Fig.5 and modified the text (lines 150-157).

2. In a previous paper, the senior author reported that recruitment of PBP1B to the division site requires PBP3 (Bertsche et al., *Mol Microbiol* 2006). This was an important enough finding to warrant mention in the abstract of that paper: “...localization of PBP1B at the septation site depended on the physical presence of PBP3, but not on the activity of PBP3.” The new manuscript under review conflicts with the earlier study. This discrepancy needs to be addressed.

Reply: We are glad that the reviewer brought up this point, which is related to the questions about FtsW asked by reviewer No 3 (see below). We did not mean to conclude that PBP1A/1B and the Z-ring membrane anchors are the only proteins present at the preseptal sites. For example, it is known that antibiotic-inhibited PBP3 remains at preseptal sites upon treatment of cells with aztreonam (Bertsche et al. 2006, *Mol Microbiol* 61, 675-690). Hence, PBP3 could be involved in recruiting PBP1A/1B to preseptal sites, which would be consistent with the interaction between PBP3 with ZipA observed in this work (Fig. 1e) and our previous finding that PBP3 interacts with PBP1B and its presence, but not activity, is required for the septal localisation of PBP1B (Bertsche et al. 2006, *Mol Microbiol* 61, 675-690). Hence, we have modified the manuscript to say that other proteins are likely present at preseptal sites, mentioning PBP3 (lines 283-285).

3. I have several concerns about how to interpret experiments showing that purified ZipA and FtsN stimulate PG synthesis by PBP1A and PBP1B in vitro. The relevant data are in Figures 3 and 4. First, it seems to me that the stimulation is very small, at best 2.6-fold (1A + ZipA in Figure 2A) but

sometimes much less, such as only 1.4 fold (1A + FtsN in Figure 2A). Moreover, this stimulation is only seen in one assay format (Figure 2) but not in two other assay formats (Figure 3). The authors have an explanation related to dimerization of the PBPs, and this might be true. But then again, it might not. This raises the question: Is a weak stimulation that is only observed under limited conditions convincing? The authors would probably say yes because BSA does not stimulate. I consider BSA a weak control. Far better would be to use mutant forms of PBPs, ZipA and FtsN to determine whether the stimulation depends on logical domains. For example, soluble ZipA should not stimulate either PBP, nor should full-length ZipA stimulate truncated PBPs that are lacking their membrane anchor domains.

Reply: BSA is often used as control protein in biochemical experiments, but the referee is right that it is not always an optimal control. We have therefore performed more experiments and found that soluble ZipA or soluble FtsN (both lacking the TM region), have no or significantly less effect on the GTase activity of PBP1A and PBP1B, confirming previous data for PBP1B and FtsN (Müller *et al.*, 2007, *JBC* 282, 36394-36402). We have added these new data into the manuscript (Figure 3 and Supplementary Fig.7; lines 163-164, 191-193 and 209-214).

It would be desirable to perform similar experiments with soluble PBP1A or PBP1B (as suggested by the referee). However, we and others found that PBP1A or PBP1B lacking the TM helix are unstable and form inclusion bodies in expression strains, and cannot be purified as soluble proteins even in the presence of detergent. The reason for this might be the close contacts between residues 83-88 of the TM region and residues 292-296 of the GTase domain of PBP1B (Sung *et al.* 2009, *PNAS* 106, 8824-8829). Presumably, these PBPs do not properly fold and precipitate when expressed without the TM region, precluding the suggested experiments.

4. Figure 3b. What do the authors make of their finding that ZipA prevents PBP3 from activating PBP1B? Since both proteins would be present in pre-septal sites *in vivo*, doesn't this finding contradict their central conclusion? In other words, the data seem to imply ZipA would only activate PBP1B when PBP3 is not present, and vice versa. But in real life both proteins would be present and cancel each other out, wouldn't they? Of course in real life FtsN would be present too, which in the *in vitro* assay overcomes the PBP3-dependent squelching of ZipA. Overall, taking the data at face value, it appears ZipA and PBP3 independently activate PBP1b, unless they are mixed, in which case they cancel each other out until FtsN comes along is very complex. The authors should comment on how they think these various results are to be interpreted.

Reply: We thank the reviewer for their excellent thoughts. Our data show that FtsN has the biggest impact on the GTase activity of PBP1B; the effects of ZipA or PBP3 are smaller. However, in the absence of knowledge about precise interaction sites and stoichiometry of complexes, and not knowing whether some interactions preclude others, any hypothesis about what happens in the cell would be highly speculative and difficult, if not impossible, to test. Moreover, in our experiments we deliberately did not add LpoB, which has the greatest stimulatory effect on PBP1B, which enabled us to delineate the effects of the other interacting proteins. Hence, we can safely conclude that the activities of PBP1B are affected by several proteins that are shown to interact with the synthase in the cell. We now make this point clearer in the revised manuscript (lines 283-285).

5. I have difficulty understanding the chromatograms shown in Figure 4a. My confusion arises from the different sizes of peaks 1 and 2. Why is peak 1 so small in the case of PBP1A alone or PBP1A + ZipA? To an outsider it looks like there was a pipetting error such that hardly any substrate was added to the reaction mixture. In the latter trace (PBP1A + ZipA), peak 2 is as large as peak 1, although both are very small. Doesn't the increase in peak 2 relative to peak 1 mean that substrate was converted to

product? If so, then ZipA stimulates PBP1A, contrary to what the authors conclude. Please clarify how these data should be read.

Reply: The reviewer made a good point about the quantity of total radioactivity eluting from the C18 column, which differs between samples in the endpoint assay with radioactive lipid II. Peak 1 is the phosphorylated disaccharide pentapeptide that originates from acid hydrolysis of unused lipid II (or ends of glycan chains carrying the undecaprenol pyrophosphate moiety). We perform this hydrolysis step after the GTase/TPase reaction because lipid II has a C55-PP moiety that precludes its elution from the C18 HPLC column used to separate the muropeptides. Boiling the sample for 10 min at pH 3-4 results in the hydrolysis of lipid II between the two phosphate groups, generating peak 1 which elutes from the C18 column. We know that the hydrolysis of lipid II is not always quantitative, giving rise to different total amount of radioactivity eluting (in particular from samples with a high amount of unused lipid II). We refrain from boiling longer or reducing the pH below 3 to increase the yield of peak 1 from lipid II hydrolysis, because this could potentially cause the loss of both phosphates, producing peak 2 without a GTase reaction. Hence, in samples with abundant unused lipid II (and only in these), the total amount of eluted radioactivity varies because peak 1 varies due to variable efficiency of lipid II hydrolysis between samples when we aim to be careful to avoid over-hydrolysis for quantification of the GTase and TPase products. We added this information to the manuscript (lines 420-427).

6. Line 125 refers to Table S3, which quantifies septal localization of PBP1A and 1B under various conditions. It looks to me like Table S3 says PBP1A localizes to division sites in 77% of wild-type cells growing normally. But I think the value should be closer to 0%. Please check this and modify the table or the text to avoid confusion.

Reply: As mentioned in our response to point 2 of reviewer 1 we defined as a ring either a sharp band that crosses the cell from side to side (closed ring) or two bright dots at both sides (open ring), consistent with the definition used in previous publications (den Blaauwen *et al.* 1999, *J Bacteriol* 181, 5167-5175; Vicente *et al.* 2006, *J Bacteriol* 188, 19-27). According to this definition, 77% of the WT cells in the absence of aztreonam show PBP1A rings. Importantly, PBP1A does not localise specifically at midcell, but localises with similar frequency all along the cell length (for comparison, PBP1B shows a much more enhanced frequency of localisation at midcell), which is consistent with previously published localisation data (Banzhaf *et al.* 2012, *Mol Microbiol* 85, 179-194). The side wall localisation of PBP1A was decreased during aztreonam incubation when PBP1A localised mainly at preseptal positions (see Figure 2 and Supplementary Fig.3). In the absence of PBP1B, PBP1A localised more frequently at midcell, as was previously described (Banzhaf *et al.* 2012, *Mol Microbiol* 85, 179-194). Hence, Supplementary Table 3 is correct and reports what the images show in Supplementary Fig.3.

Minor points

7. Line 63. The passage is written in a way that implies FtsW is a lipid II flippase but not a transglycosylase, but this is a matter of contention. It would be better to say that FtsW is proposed by some to be a lipid II flippase but by others to be a PG transglycosylase.

Reply: We agree with the reviewer and changed the text as suggested (lines 62-65).

8. Line 139. Here the text states that PBP1A and PBP1B are recruited to division sites "...presumably linked to the Z-ring by ZipA, although FtsN may also contribute to their recruitment." As noted above, what about the previous claim that localization of PBP1B required the physical presence of PBP3?

Reply: Because ZipA-depleted cells showed PBP1A/PBP1B along the cell length instead of at preseptal sites (Supplementary Fig.5) we suggested that ZipA is the main protein responsible for recruitment of these PBPs. However, we changed the text to say that other proteins may be involved in the recruitment of PBP1A and PBP1B (lines 147-149).

9. Related to the PBP1b-PBP3 interaction, I think the manuscript under review brings to 8 the number of proteins the senior author has reported bind directly to PBP1B: Itself (dimerization), LpoB, MltA, MipA, PBP3, FtsN, FtsW and now ZipA. I think this should be noted somewhere. First, a global picture of PBP1b's reported interaction partners provides important context for thinking about how PBP1b might be recruited and regulated *in vivo*.

Reply: Indeed, as the reviewer pointed out, we have learned in the recent years that PBP1B interacts with several other proteins (the reviewer mentioned MltA, but this hydrolase does not directly interact with PBP1B); the latest addition is the C55-PP pyrophosphatase PgpB (Hernandez-Rocamora *et al.*, 2018, *Cell Surf* 2, 1-13). Although we do not know if all these interactions occur at all times in the cell (which might be unlikely) and we know the site of interaction only for few proteins, we hypothesize that the synthase does not function in isolation in the cell, but its activity and site of activity is controlled by multiple interaction partners. As suggested, we now noted these interactions in the revised manuscript, and we speculate that it is unlikely that in the cell all of these interactions occur at all times with all molecules of PBP1B (lines 300-304).

Second, the fact that PBP1b has some many binding partners means some assays in the manuscript under review are missing components that a likely to be relevant. This could affect the results of binding and PG synthesis assays.

Reply: We agree with the reviewer and don't discard the possibility that other binding partners might affect the outcomes of interaction or activity assays with PBP1B. Indeed, as we mentioned in our reply to point 4, we deliberately did not add LpoB, the protein with the greatest stimulatory effect on PBP1B, enabling us to delineate the effects of other interacting proteins. We added this point to the discussion (Lines 283-285 and 301-304).

Third, readers will realize that the interaction assays used in this paper to show ZipA binds to PBP1b have a history of finding many such interactions, which could mean PBP1b is part of a large complex or that the assays are prone to false positives. I don't think this manuscript is the place to take up the meaning of such a large number of interaction partners, but I do think the existence of the growing set of interactions should be noted. (Refs for interactions: Laclercq *et al.*, 2017; Typas *et al.*, 2010; Muller *et al.*, 2007; Bertsche *et al.*, 2006; Bertsche *et al.*, 2005; Vollmer *et al.*, 1999)

Reply: As mentioned above, it might be unlikely that all of the interactions we measure *in vitro* and detect *in vivo* occur constantly in the cell; the situation is presumably much more dynamic and, for example, different interactions might prevail during cell elongation or cell division, or under different growth conditions. All interactions have been verified by different assays *in vitro* and in the cell (cross-linking), hence either all assays are prone to false-positive or interactions are real. We now mention the interactions of PBP1B in the revised manuscript (lines 300-304).

10. Line 79. Do the authors mean to include RodA as division protein?

Reply: We agree with the reviewer that the sentence was confusing and corrected it (line 78-80).

11. Line 249. "If zipA is absent, FtsN becomes the only linker between the cytosolic proto-ring (formed by FtsA* and FtsZ) and the PG synthases..." If that's the model, why does Figure 1a show PBP3 bound directly to FtsA? Maybe the passage should specify "bifunctional" PG synthases.

Reply: We thank the reviewer for this point and have specified “bifunctional PG synthases” (line 277).

12. Line 145 and Figure 3. ZipA increases GTase activity slightly when added to PBP1A or 1B in vitro. The authors interpret this to mean ZipA binds to and activates both proteins. This is surprising and interesting because the two PBPs do not have homologous membrane anchors. The text should note this. Or do the authors suspect that ZipA interacts directly with the GTase domains? These are at least homologous.

Reply: The stimulatory effect of ZipA was stronger for PBP1A and weaker for PBP1B, so there is a difference between the PBPs. Without knowing the binding interface, we refrain from speculating too much about the degree of homology of the interaction sites on the PBPs.

13. Please revise the part of the methods section that deals with strain construction to make it easier for the reader to understand. I had to constantly bounce back and forth between the text and the strain table, which was in a separate document. For example, line 277 now reads: To produce strain MPW22, WM2935 was transformed with pCH32 and transduced with the P1 lysate from the strain WM1304 (Δ zipA::aph). But what are MPW22, pCH32 and MW2935? The sentence would be easy to follow if key genotypes were included in the text, e.g., “To produce strain MPW22 [ftsA E124A zipA::aph/pCH32(zipA, ftsZ)], WM2935 (ftsA E124A) was transformed with pCH32 (zipA, ftsZ) and transduced with the P1 lysate from the strain WM1304 (Δ zipA::aph).”

Reply: Many thanks for this point, we added the genotypes into the text (lines 317-326) and Figure 5.

14. Line 296. Was the entire construct sequenced? I assume only the “insert” was sequenced even though the entire plasmid was made by PCR. Please clarify.

Reply: Indeed, only the insert was sequenced and this was corrected in the manuscript (line 347).

15. The Lpo overproduction plasmids are pET28 derivatives but Table S2 indicates the His tag is at the N-terminus. Is that correct?

Reply: That is correct.

Please check other plasmids and figures for accuracy because Table S2 lists pET28 derivatives that produce ZipA-His or soluble ZipA-His, but in Figure 1 there is His-ZipA but sZipA-His. There might not be any mistakes here because a variety of overproduction plasmids were used in this study.

Reply: There is no mistake about the position of the His-tags.

Reviewer #3 (Remarks to the Author):

This manuscript investigates a somewhat mysterious process that has been referred to as PIPS, which is an FtsZ dependent, but PBP3 independent, incorporation of DAP at the division site. A similar process has been described in *Caulobacter* but it's too early to say whether it is the same. In this paper, the authors investigate the requirements for this process in *E. coli*. In part they are guided by a paper from Young's lab that shows that PIPS depends upon ZipA but can also occur in FtsA* strains that bypass ZipA. They first detect interactions between PBP1B and ZipA, PBP1A and ZipA as well as PBP3 and ZipA by chemical crosslinking and IP, but not with any of these with FtsN. These interactions are by testing direct interactions using purified proteins and a pull down assay. They then show by one assay that the GTase, but not the TPase, activities of PBP1A and 1B are stimulated in vitro. 1A is stimulated 2X by ZipA and marginally by FtsN whereas 1B is stimulated by FtsN and

marginally by ZipA. In another assay (lower protein concentration) 1A is not affected whereas 1B is stimulated by FtsN. They then show that 1A and 1B are localized to the division site in the presence of aztreonam. This is the most dramatic localization of these proteins to potential division sites that I have seen. Also, they fuse HADA labelling to show that PIPS occurs in a *ftsA** strain if FtsN or ZipA is present but not if both are absent. They then suggest that PIPS is due to 1A or 1B being recruited to the division site by ZipA or FtsN and being stimulated there.

One of my main concerns with the paper is there is what is not mentioned. First of all there is no mention of FtsW as a potential polymerase. Instead it is dismissed based upon the failure of one lab to observe activity in an *in vitro* assay.

Reply: We did not mention the proposed GTase activity of FtsW because this is beyond the topic of our manuscript. It is indeed possible that some laboratories purified inactive versions of FtsW proteins or that FtsW needs an activator. We believe that more experimental evidence is needed to prove that FtsW is a GTase, which is not the topic of this manuscript. Hence, we added that FtsW was proposed to be a GTase, with references to Meeske *et al.* 2016, *Nature* 537, 634-638 and Emami *et al.* 2017, *Nature Microbiology* 2, 16253 (lines 62-65).

No mention is made of the fact that a close paralog of FtsW, RodA has been shown to be a polymerase by one lab (Meeske *et al.* 2016) and inferred to be a polymerase by another (Emami *et al.* 2017).

Reply: As suggested we added the references mentioned by the reviewer (lines 62-65).

Thus, it is possible that the PIPS seen in the absence of ZipA is due to FtsW (it should be localized).

Reply: Whilst we cannot formally exclude this possibility, at this time we find it rather speculative because (i) to our knowledge there is no data showing that FtsW can synthesize PG and (ii) the crystal structure of its paralogue, RodA from *T. thermophilus*, does not show the two substrate binding sites that would be expected to be present in a GTase: one for lipid II and another for the growing glycan strand. Importantly, it is known that FtsK and FtsQLB are necessary to position FtsW at midcell (Mercer and Weiss 2002, *J Bacteriol* 184, 904-912; Goehring *et al.* 2005, *Genes Dev* 19, 127-137; Goehring *et al.* 2006, *Mol Microbiol* 61, 33-45), and preseptal PG synthesis takes place in cells depleted of FtsK or FtsQ (Potluri *et al.* 2012, *J Bacteriol* 194, 5334-5342), suggesting that preseptal PG synthesis does not require the localisation of FtsW at preseptal positions. Hence, we refrained from localising FtsW but refer to the above mentioned references, adding statement that preseptal PG synthesis does not appear to require the localisation of FtsW to preseptal sites (lines 285-288).

It is also possible that PIPS seen in the absence of FtsN is also due to FtsW. Of course some other protein must do the crosslinking. Second, a paper by the Bernhardt lab shows that proteins from *Pseudomonas*, PBP1B and a protein that substitutes for LpoB, work in *E. coli* and can take the place of PBP1B and LpoB. It is unlikely that the proteins from this organism would retain the specific contacts with the division proteins from *E. coli*.

Reply: See previous point regarding PIPS due to FtsW. The referee made a valid point that it is difficult to understand why PBP1B from *Pseudomonas* is functional in *E. coli*. However, we would like to make two points: First, PBP1B from *E. coli* and *Pseudomonas* show significant degrees of identity and similarity. For example, the 24/23 amino acid long TM helix of both proteins contains 11 identical and 4 similar residues. Hence, it is possible that (part of the) interaction site is conserved or similar. Second, the complementation experiments published by the Bernhardt lab (Greene *et al.* 2018, *PNAS* 115, 3150-3155) did not use an *E. coli* strain in which the native PBP1B gene was replaced by the PBP1B gene from *Pseudomonas*, which would express the *Pseudomonas* PBP1B at

similar level as the *E. coli* protein. Rather, they overexpressed the gene encoding *Pseudomonas* PBP1B from a multicopy plasmid under the control of an IPTG inducible promoter. Figure S10 of Greene *et al.* shows that the *Pseudomonas* PBP1B was massively (>100-fold) overproduced in *E. coli* compared to the levels of native *E. coli* PBPs or induced *E. coli* PBP1B. We hypothesize that this massive overproduction might render PBP1B from *Pseudomonas* functional in *E. coli* despite it might not engage in all protein-protein interactions that *E. coli* PBP1B is involved with.

The crosslinking of ZipA with 1A, 1B and PBP3 is surprising for several reasons. One is very little of ZipA is in the periplasm so the crosslinking must occur in the cytoplasm. The crosslinking reagent is for primary amines and ZipA only has no lysines in the periplasm and none in the transmembrane region.

Reply: We thank the reviewer for these points which we would like to clarify. First, ZipA has its N-terminus on the periplasmic side of the membrane, hence it has one primary amino group (present in the first amino acid) in the periplasm. Second, in our experiments we used DSP as cross-linker which is known to be able to diffuse across the cell membrane (Xiang *et al.* 2004, *Nucleic Acids Res.* 32, e185), allowing also for cross-linking of cytosolic amino groups. We added a sentence to the manuscript to mention that the crosslinker can diffuse across the cell membrane (lines 397-399).

The authors suggest the transmembrane region is responsible for interactions between ZipA and its partners. It is not clear how this would stimulate the enzyme activity of 1A since the enzymatic domains are not near the membrane.

Reply: The reviewer is right that we currently do not know the mechanism of activation. However, the TM helix of PBP1B is close to the GTase domain and any interaction in this region could potentially affect the GTase rate. The TPase is further away, but we previously published that the TPase activity depends on ongoing GTase reactions, indicating that both activities are coupled (Bertsche *et al.* 2005, *JBC* 280, 38096-38101; Born *et al.* 2006, *JBC* 281, 26985-26993). Alternatively, or in addition, the interaction with ZipA could stabilize the dimeric form of PBP1B, which has been shown to be more active than the monomeric form (Bertsche *et al.* 2005, *JBC* 280, 38096-38101). In the manuscript, we refrained from speculating about the possible mechanisms of stimulation as the effects are not as big as, for example, with LpoB.

The localization of 1A and 1B is the most dramatic I've seen. It would be nice to show that this coincides with the Z ring. What is surprising is that 1A displays better localization than 1B, which has been linked to division in the past.

Reply: Please see our responses to point 2 of reviewer 1 and point 6 of reviewer 2. We do not show that PBP1A localised better than PBP1B (Supplementary Table 3). The portion of cells showing ring-like structures was higher when PBP1B is labelled, with the exception of the condition "40 min aztreonam" comparing PBP1B localisation in $\Delta mrcA$ cells (71%) and PBP1A localisation in $\Delta mrcB$ cells (73%).

I worry about experiments that depend so much on biochemistry and have no mutants to support the work to show the physiological significance.

Reply: We agree with the reviewer that biochemical and physiological data should complement each other to obtain meaningful insights about cellular processes. We would hope that the reviewer sees that our biochemical data align well with the cellular localisation of proteins and phenotypes of depletion strains, showing that membrane anchors link PG synthases and the FtsZ cytoskeleton at the pre-divisional phase in the cell cycle.

Reviewers' comments:

Reviewer #1 (Remarks to the Author):

In their revised manuscript, Pazos et al have addressed most of the points raised in the original review of their work investigating the functional overlap between ZipA and FtsN-FtsA in stimulating pre-septal cell wall synthesis in *E. coli*.

They propose a model that is consistent with and supported by their data wherein ZipA and FtsN-FtsA play redundant roles in promoting PBP1AB-mediated cell wall synthesis prior to constriction. I am still not completely convinced that pre-septal cell wall synthesis is or should be an essential activity (Lines 284-294, Why does the cell need to "hand off" machinery from elongation to constriction? What are the limiting factors that wouldn't be efficiently engaged/localized/active at the division site in the absence of PIPS?), and it still seems possible that ZipA and FtsN-FtsA play a redundant role in constriction, itself.

Other specific points:

Line 53, line 64 and elsewhere: There are now additional reports of in vitro GTase activity of *S. thermophilus*, *S. aureus*, and *P. aeruginosa* FtsW (Taguchi et al bioRxiv 2018) and of *E. coli* RodA (Rohs et al bioRxiv 2018). It is important to therefore give more credence/discussion to the possibility of FtsW and/or RodA playing a role in both pre-septal PG synthesis and PG synthesis for division that might explain some of their results.

Lines 112-119: The authors put in admirable effort to address this point, and the new data do support their model. However, there is still the caveat that the His tag is in a different place (N-ter vs C-ter, at least as annotated in the figures) for the full length vs sZipA and W-ZipA. The tag at one terminus vs the other could interfere with interaction and/or pulldown.

The "interaction" with PBP3 and PBP1A looks very weak in 1% Triton (it looks similar to the sZipA pulled down with His-PBP3 in Fig 1e). Is there a difference in reaction conditions between Fig 1e and Supp Fig 1b? If not, is the relatively weak pulldown of ZipA in Fig S1b just variability in the assay? If the conditions are different, what are the conditions used for Fig 1e? Given the strong pulldown of ZipA with FtsN at lower Triton concentration, a clear delineation of why the pulldowns are interpreted the way they are (other than consistency with the co-IP data) is warranted.

Line 164: My take home here from just looking at the data would be that (1) ZipA stimulates PBP1A GTase with a minor additive effect with FtsN and (2) FtsN stimulates PBP1B GTase with no additive effect of ZipA. FtsN doesn't do much to stimulate 1A and ZipA doesn't do much to stimulate 1B. However, here and elsewhere the stimulation of PBP1A and PBP1B by FtsN and ZipA are treated as somewhat equivalent (i.e. FtsN and ZipA each stimulate both PBP1A and PBPB). It seems likely there's more specificity to their interactions than that, which should be discussed.

Minor corrections:

Line 32: "zipA" should be lowercase (even at the beginning of a sentence) since it refers to a gene

Line 33: Genes aren't depleted, transcripts or proteins are.

Line 89: should be "...that ZipA interacts with and enhances the GTase activity..."

Line 129: For clarity, consider "...exponentially growing cells using specific antibodies before and after the inhibition of cell division..."

Reviewer #2 (Remarks to the Author):

The authors have revised their manuscript and addressed my major concerns. I especially appreciate the addition of experiments showing soluble versions of ZipA and FtsN do not stimulate the activity of PBP1a/1b very much, if at all. These new data strengthen one of the central claims of the paper.

I am still perplexed about how to integrate the new findings with their older report that septal localization of PBP1b "depends upon the presence of PBP3 at [the division site]" (Bertsche et al., 2006). The authors revised their manuscript to say that "other cell division proteins" besides ZipA "may contribute" to recruitment (line 152). This wording seems overly tentative and unnecessarily vague if PBP3 is REQUIRED for recruitment of PBP1b, by their own hand, no less. But I do not wish to contest this point further. The newly identified ZipA-PBP interactions are interesting and potentially important. Time will tell how they fit into the larger picture of multiprotein interactions during *E. coli* division.

Reviewer #3 (Remarks to the Author):

The authors have been quite responsive to the previous reviews. The authors argue that PIPS can not be due to FtsW since PIPS occurs in *ftsQ(Ts)* mutants that should not have FtsW localized. That is a strong argument however, I am not sure that is shown here. In Fig. 5 the authors present an *in vivo* experiment to look at PIPS. In this experiment the authors deplete zipA or ftsN or both in the *FtsA-E124A* background. In the text the authors (lines 231-to end of results) state that *ftsA-E124A* suppresses loss of zipA or ftsN but not loss of both. Although loss of zipA is readily suppressed the loss of FtsN is not completely suppressed (Fig. 5A) and this needs to be pointed out. Although *ftsA-E124A* was originally reported to suppress the loss of ftsN, a paper from the de Boer lab showed that to suppress the loss of FtsN, *ftsA-E124A* had to be over expressed (Girding et al. 2009). The result in 5A is consistent with these results and also with the report that the *ftsA-E124A* allele on the chromosome was shown to not bypass FtsN although less FtsN was required for survival (Du et al, PNAS 113: E5052[2016]). The *ftsA-E124A* allele helps some with the loss of FtsN (however, colony formation is still down about 1000 fold in the FtsN depletion so it does not completely suppress [Fig. 5]). As a result my worry is that with the approach used here FtsN may not be sufficiently depleted. I worry that the so called PIPS in Fig. 5f may actually be due to residual FtsN which can back recruit FtsW to do transglycosylation (still need a transpeptidase which may be 1A or 1B).

However, the absence of PIPS in *ftsQ(Ts)* strain makes FtsW unlikely.

It would be nice for the authors to show that Z rings are still present in Fig. 5G. To me it is not clear why depletion of zipA and FtsN is worse than depletion of both and it has not been shown that Z rings are still present under such conditions although the authors assume they are.

Rebuttal - Reviewer's comments in blue text; our reply in black text

Reviewer #1 (Remarks to the Author):

In their revised manuscript, Pazos et al have addressed most of the points raised in the original review of their work investigating the functional overlap between ZipA and FtsN-FtsA in stimulating pre-septal cell wall synthesis in *E. coli*.

They propose a model that is consistent with and supported by their data wherein ZipA and FtsN-FtsA play redundant roles in promoting PBP1AB-mediated cell wall synthesis prior to constriction. I am still not completely convinced that pre-septal cell wall synthesis is or should be an essential activity (Lines 284-294, Why does the cell need to "hand off" machinery from elongation to constriction? What are the limiting factors that wouldn't be efficiently engaged/localized/active at the division site in the absence of PIPS?), and it still seems possible that ZipA and FtsN-FtsA play a redundant role in constriction, itself.

Reply: We thank the reviewer for commenting that our model is consistent with the data and for asking these insightful questions which we are asking ourselves. Indeed, we cannot know why this particular mechanism has evolved and the cell needs to "hand off" machineries (although potentially one could envision a different mechanism). Regarding the question about the limiting factors, our data suggest that the link between the FtsZ-ring and the PG synthases (PBP1A and PBP1B) is key for pre-septal PG synthesis, which appears to be required for the following constriction. Fenton *et al.* showed that an impaired interaction between MreB and FtsZ leads to the mislocalization of PBP2 and PBP1B, and the absence of pre-septal and septal PG synthesis, suggesting a defective transfer of the PG synthesis machinery from the elongasome to the divisome (Fenton *et al.* 2013 *EMBO J* 32, 1953-1965). We mention and discuss these findings (lines 281-286).

As further suggested by the reviewer, we discuss that ZipA and FtsN-FtsA are also required for constriction itself (lines 270-272).

Other specific points:

Line 53, line 64 and elsewhere: There are now additional reports of *in vitro* GTase activity of *S. thermophilus*, *S. aureus*, and *P. aeruginosa* FtsW (Taguchi et al bioRxiv 2018) and of *E. coli* RodA (Rohs et al bioRxiv 2018). It is important to therefore give more credence/discussion to the possibility of FtsW and/or RodA playing a role in both pre-septal PG synthesis and PG synthesis for division that might explain some of their results.

Reply: We thank the reviewer for pointing to these archived manuscripts, which were not published at the time of submission of our revised manuscript. We respectfully wish to correct a statement made by the reviewer: Rohs et al 2018 did not report *in vitro* GTase activity of *E. coli* RodA but they reported GTase activity of a RodA-PBP2 fusion protein. We have now added references to Taguchi et al. 2018 and Rohs et al. 2018, to the Introduction, mentioning the published GTase activity of RodA and FtsW versions/fusions (lines 63-65). However, it is important to note that previous work by others showed that pre-septal PG synthesis does occur in the absence of RodA, FtsK or FtsQ (Potluri *et al.* 2012, *J Bacteriol* 194, 5334-5342). That pre-septal PG synthesis occurs in the absence of FtsK and FtsQ also discards any role of FtsW, as it is not recruited to pre-septal sites under these conditions.

Lines 112-119: The authors put in admirable effort to address this point, and the new data do support their model. However, there is still the caveat that the His tag is in a different place (N-ter vs C-ter, at least as annotated in the figures) for the full length vs sZipA and W-ZipA. The tag at one terminus vs the other could interfere with interaction and/or pulldown.

Reply: We thank the reviewer for acknowledging our effort. We would like to explain why the position of the His-tags does not matter: (i) N-terminal His-tag. We found that His-ZipA (full-length) does interact with PBP1A and PBP1B. If this were a false-positive interaction caused by the N-terminal His-tag, then the tag-less ZipA (full-length) should not stimulate the PBPs (which it does). The tag-less sZipA (without the TM region) does not stimulate the PBPs; hence, the effects we see depend on the TM region and are independent of the N-terminal His-tag. Another example is His-PBP3 which only interacts with full-length ZipA but not with sZipA or WALP-ZipA showing that the

N-terminal His-tag of PBP3 does not generally interact with all ZipA versions (ii) C-terminal His-tag. We found that sZipA-His or WALP-ZipA-His do not interact with PBP1A or PBP1B. If this were false-negative results caused by the C-terminal His-tag preventing an interaction then the same effect would be expected for FtsN-His (C-terminal His-tag). However, we found that FtsN-His stimulates PBP1B, showing that C-terminal His-tags do not generally prevent interactions with PBPs. This is further supported by the fact that sFtsN-His does not stimulate PBP1B, showing that the stimulation depends in the N-terminal region and is independent of the C-terminal His-tag.

The "interaction" with PBP3 and PBP1A looks very weak in 1% Triton (it looks similar to the sZipA pulled down with His-PBP3 in Fig 1e). Is there a difference in reaction conditions between Fig 1e and Supp Fig 1b?

Reply: Yes, the concentration of detergent was different: 0.05% Triton X-100 (Figure 1E) and 1% Triton X-100 (Supp Fig 1b). We clarified this in the legend of Figure 1.

If not, is the relatively weak pulldown of ZipA in Fig S1b just variability in the assay?

Reply: As we mentioned in the previous rebuttal letter, the amount of His-ZipA is only lower in the elution of the experiment with PBP1A. We do not know the reason for these differences, but they might be due to different strength of binding to Ni-NTA beads between sZipA-His and His-ZipA.

If the conditions are different, what are the conditions used for Fig 1e?

Reply: The concentration of Triton X-100 is 0.05%; this is mentioned in the legend of Fig. 1e.

Given the strong pulldown of ZipA with FtsN at lower Triton concentration, a clear delineation of why the pulldowns are interpreted the way they are (other than consistency with the co-IP data) is warranted.

Reply: We qualitatively compare the results from each pair of proteins. In the case of PBP1A, the pulldown experiments in the presence of 0.05% Triton (Fig 1c, top panel) or 1% Triton (Supp Fig 1b, top panel) show a similar band intensity in the elution sample. The same is true for the pulldown experiments with PBP1B (Fig 1d, top panel; and Supp Fig 1b, top panel). Our interpretation is that in these cases the proteins do interact with each other and the interaction is not affected by the high concentration of detergent. In the case of FtsN-His + ZipA sample, there is a similar band intensity of both proteins in the elution sample when the buffer contained 0.05% Triton (Supp Fig 1a, top panel), but the band intensity of ZipA is significantly lower than that of FtsN-His when the buffer contained 1% Triton (Supp Fig 1b, bottom panel). We concluded that the ZipA-FtsN-His interaction is sensitive to detergent and likely unspecific. Unspecific interactions are often mediated by surface-exposed hydrophobic regions in proteins. In the case of ZipA and FtsN-His, the most hydrophobic regions are the TM regions which might interact unspecifically with each other in the presence of 0.05% Triton, but not in the presence of 1% Triton or in the natural membrane environment. As the reviewer mentioned, this interpretation is consistent with the co-IP data showing a lack of interaction between ZipA and FtsN in the cell. We discuss the Triton effect on the ZipA-FtsN-His interaction on lines 118-122.

Line 164: My take home here from just looking at the data would be that (1) ZipA stimulates PBP1A GTase with a minor additive effect with FtsN and (2) FtsN stimulates PBP1B GTase with no additive effect of ZipA.

Reply: We agree with the reviewer, and so it was written in the Results section (lines 165-168) and Discussion section (295-297).

FtsN doesn't do much to stimulate 1A and ZipA doesn't do much to stimulate 1B. However, here and elsewhere the stimulation of PBP1A and PBP1B by FtsN and ZipA are treated as somewhat equivalent (i.e. FtsN and ZipA each stimulate both PBP1A and PBP1B). It seems likely there's more specificity to their interactions than that, which should be discussed.

Reply: We thank the reviewer for this point, which we now make clearer in the manuscript. We address this point in the discussion (lines 300-302), where we mentioned the effect of FtsN on the activities of PBP1B and we modified the text (lines 29, 88-91 and 160) to make sure to mention that

the effects are not equivalent.

Minor corrections:

Line 32: “zipA” should be lowercase (even at the beginning of a sentence) since it refers to a gene

Reply: We thank the referee for this suggestion and corrected the text.

Line 33: Genes aren't depleted, transcripts or proteins are.

Reply: Many thanks, we corrected the text.

Line 89: should be “...that ZipA interacts with and enhances the GTase activity...”

Reply: We changed the text taking into account the previous point about the specificity of the effects on PBP1A and PBP1B activity.

Line 129: For clarity, consider “...exponentially growing cells using specific antibodies before and after the inhibition of cell division...”

Reply: We thank the referee, we clarified the text as suggested.

Reviewer #2 (Remarks to the Author):

The authors have revised their manuscript and addressed my major concerns. I especially appreciate the addition of experiments showing soluble versions of ZipA and FtsN do not stimulate the activity of PBP1a/1b very much, if at all. These new data strengthen one of the central claims of the paper.

Reply: We thank the reviewer for their positive comments.

I am still perplexed about how to integrate the new findings with their older report that septal localization of PBP1b "depends upon the presence of PBP3 at [the division site]" (Bertsche et al., 2006). The authors revised their manuscript to say that "other cell division proteins" besides ZipA "may contribute" to recruitment (line 152). This wording seems overly tentative and unnecessarily vague if PBP3 is REQUIRED for recruitment of PBP1b, by their own hand, no less. But I do not wish to contest this point further. The newly identified ZipA-PBP interactions are interesting and potentially important. Time will tell how they fit into the larger picture of multiprotein interactions during *E. coli* division.

Reply: It is known that the activity or presence of PBP3 is not required for preseptal PG synthesis, which occurs in *ftsQ(ts)* or *ftsK(ts)* strains when PBP3 is not recruited to division sites under non-permissive condition (Potluri *et al.* 2012, *J Bacteriol* 194, 5334-5342). Under standard growth condition septation takes place immediately after PIPS, so PBP3 is likely recruited to the division site during preseptal PG synthesis. It is possible that the presence of PBP3 at the division site (although not yet active) might help or enhance the recruitment or stabilization of PBP1B during preseptal PG synthesis. This point, including FtsK, FtsQ and FtsEX, was already discussed in the ms (lines 286-288).

Reviewer #3 (Remarks to the Author):

The authors have been quite responsive to the previous reviews. The authors argue that PIPS can not be due to FtsW since PIPS occurs in *ftsQ(Ts)* mutants that should not have FtsW localized. That is a strong argument however, I am not sure that is shown here. In Fig. 5 the authors present an *in vivo* experiment to look at PIPS. In this experiment the authors deplete zipA or ftsN or both in the FtsA-E124A background. In the text the authors (lines 231-to end of results) state that *ftsA-E124A* suppresses loss of zipA or ftsN but not loss of both. Although loss of zipA is readily suppressed the loss of FtsN is not completely suppressed (Fig. 5A) and this needs to be pointed out. Although *ftsA-E124A* was originally reported to suppress the loss of ftsN, a paper from the de Boer lab showed that to suppress the loss of FtsN, *ftsA-E124A* had to be over expressed (Girding et al. 2009). The result in

5A is consistent with these results and also with the report that the *ftsA*-E124A allele on the chromosome was shown to not bypass FtsN although less FtsN was required for survival (Du et al, PNAS 113:E5052[2016]). The *ftsA*-E124A allele helps some with the loss of FtsN (however, colony formation is still down about 1000 fold in the FtsN depletion so it does not completely suppress [Fig. 5]).

Reply: We thank the reviewer for these thoughtful comments and for agreeing that FtsW is not required for pre-septal PG synthesis. We modified the text and added quotations to the references mentioned by the reviewer (line 230).

As a result my worry is that with the approach used here FtsN may not be sufficiently depleted. I worry that the so called PIPS in Fig. 5f may actually be due to residual FtsN which can back recruit FtsW to do transglycosylation (still need a transpeptidase which may be 1A or 1B). However, the absence of PIPS in *ftsQ*(Ts) strain makes FtsW unlikely.

Reply: The MPW23 cells (Fig. 5f) perform preseptal PG synthesis because they contain ZipA. FtsN was depleted using the same depletion system and condition than in the MPW29 cells shown in Fig. 5g (160 min growth at non-permissive temperature). We know that the level of FtsN is sufficiently decreased in MPW29 cells (Fig. 5g) to decrease preseptal PG synthesis to a similar extent as in *ftsA** Δ *ftsN* cells depleted of ZipA (MPW30, Supplementary Table 1). This suggests that the FtsN depletion system reduces FtsN to a level that does not allow the back-recruitment of proteins for preseptal PG synthesis (PBP1A and PBP1B).

We agree with the reviewer that FtsW is unlikely to be involved in preseptal PG synthesis. FtsW is not recruited to preseptal PG synthesis sites present in *ftsQ*(ts) or *ftsK*(ts) strains (Mercer and Weiss 2002, *J Bacteriol* 184, 904-912; Goehring *et al.* 2005, *Genes Dev* 19, 127-137; Goehring *et al.* 2006, *Mol Microbiol* 61, 33-45; Potluri *et al.* 2012, *J Bacteriol* 194, 5334-5342).

It would be nice for the authors to show that Z rings are still present in Fig. 5G. To me it is not clear why depletion of *zipA* and FtsN is worse than depletion of both and it has not been shown that Z rings are still present under such conditions although the authors assume they are.

Reply: We thank the reviewer for suggesting this experiment. We have now localized FtsZ in cells depleted of ZipA and FtsN. As expected, FtsZ does localize in regular ring pattern in these filamentous cells (most likely by its membrane anchor FtsA). We added this finding as new Supplementary Figure 8 and mention the information in the text (line 237-239).

Why is the depletion of ZipA and FtsN together worse than the depletion of only one of these proteins? According to our model (Figure 6) both ZipA and FtsA-FtsN maintain the link between the FtsZ-ring and PBP1A/1B, consistent with the ability of FtsA* to bypass each single mutant (Δ *zipA* or Δ *ftsN*). Depleting both, FtsN and ZipA abolishes the link between FtsZ and PBP1A/1B, resulting in non-viable cells that are unable to perform preseptal PG synthesis. As ZipA and FtsA-FtsN localise at division sites until septation is finished, we discuss that this “linker” function extends from preseptal PG synthesis into septum synthesis (lines 270-272).

REVIEWERS' COMMENTS:

Reviewer #1 (Remarks to the Author):

The authors have addressed my comments from the prior rounds of review. One final note, the title does not seem consistent with the content of the paper. Specifically, the phrase "... to initiate bacterial cell division" (which I would associate with a role in initiating the constriction phase) isn't addressed/supported by the data in this paper. "... to direct pre-septal cell wall synthesis" is more accurate.

Reviewer #3 (Remarks to the Author):

The authors have been very responsive and answered my questions as well as those of other reviewers. The work is quite challenging and the authors have put enormous effort into this project. The new fig. S8 answered my main concern.

Reviewers' points in blue text; our response in black text.

REVIEWERS' COMMENTS:

Reviewer #1 (Remarks to the Author):

The authors have addressed my comments from the prior rounds of review. One final note, the title does not seem consistent with the content of the paper. Specifically, the phrase "... to initiate bacterial cell division" (which I would associate with a role in initiating the constriction phase) isn't addressed/supported by the data in this paper. "... to direct pre-septal cell wall synthesis" is more accurate.

Reply: We appreciate the suggestion, however it is generally accepted that cell division does not start with cell constriction. There is an early stage in which FtsZ, FtsA and ZipA are recruited to midcell and no constriction or septation is visible. So we do think that the current title is accurate.

Reviewer #3 (Remarks to the Author):

The authors have been very responsive and answered my questions as well as those of other reviewers. The work is quite challenging and the authors have put enormous effort into this project. The new fig. S8 answered my main concern.

Reply: We thank the reviewer for their positive comments.